# Decoding-based Regression

**Xingyou Song**[*], **Dara Bahri**[*]

**Google DeepMind**
[*]Equal Contribution.

Reviewed on OpenReview: `https://openreview.net/forum?id=avUQ8jguxg`

## Abstract

Language models have recently been shown capable of performing regression wherein numeric predictions are represented as decoded strings. In this work, we provide theoretical grounds for this capability and furthermore investigate the utility of causal sequence decoding models as numeric regression heads given any feature representation. We find that, despite being trained in the usual way - for next-token prediction via cross-entropy loss - decoder-based heads are as performant as standard pointwise heads when benchmarked over standard regression tasks, while being flexible enough to capture smooth numeric distributions, such as in the task of density estimation.

## 1 Introduction

Despite being originally developed for the purpose of text generation and chat applications, large language models (LLMs) have recently been applied for new applications, particularly one of which is regression, and more broadly the prediction of numeric outcomes. Vacareanu et al. (2024) have shown service-based LLMs such as ChatGPT and Gemini capable of regression with performance comparable to that of traditional regression methods such as random forests, while Song et al. (2024); Akhauri et al. (2025) have shown smaller custom language models can be trained specifically on multiple regression tasks for transfer learning.

For an input-output pair $(x, y)$, where $x$ is a feature vector and $y$ is a real number, a regression model's performance is determined by two factors: how it processes $x$ and how its output "head" represents $y$. While the mentioned works (Vacareanu et al., 2024; Song et al., 2024; Akhauri et al., 2025) can be seen as *text-to-text regression* where both $x$ and

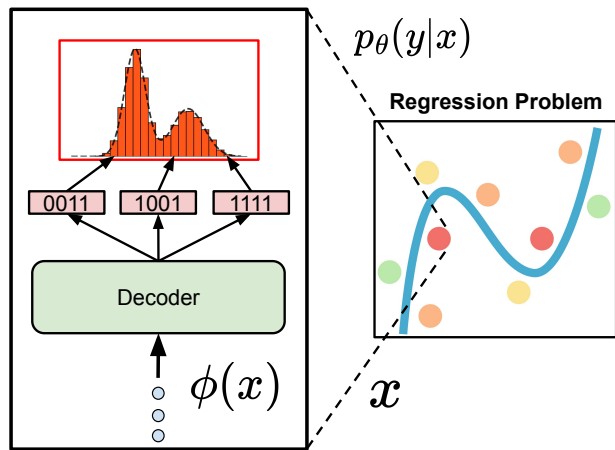

Figure 1: Given any feature representation $\phi(x)$, we attach a decoding-based head to output predictive distribution $p_\theta(y|x)$.

$y$ are represented as tokens, this combination is not necessarily required. Tang et al. (2024) investigate the isolated case where LLM embeddings of $x$ are attached to feed-forward networks as regression heads, while Nguyen et al. (2024) investigate the case when these embeddings are eventually attached to Gaussian distribution heads. Both can be seen as particular instances when $x$ is represented as text, while common regression heads are still used. However, there has not been work investigating the inverse situation, i.e. $y$ is represented as text or structured tokens. One could do so by using decoding-based regression heads, where for example, tokens `<1><.><2><3>` can be decoded to represent 1.23, a technique used in several works

training language models for specific numeric tasks, such as arithmetic (Nogueira et al., 2021), linear algebra (Charton, 2022), and symbolic regression (d'Ascoli et al., 2022).

In contrast to traditional deterministic or parametric distribution (e.g. Gaussian) regression heads, decoding-based heads may be much more flexible, as they can represent any numeric distribution approximately over $\mathbb{R}$ without the need for explicit normalization. However, due to their initial lack of inductive bias over numeric distances, numeric token representations and sequential dependencies may need to be *learned* using additional training data, and thus it is worth empirically investigating these trade-offs in isolated, controlled experiments. Our research provides valuable insights on using numbers as an output modality of token-based autoregressive decoders. Our specific contributions and findings are thus as follows:

- We formalize decoding-based regression, i.e. explicitly define tokenization schemes for numbers, establish training and inference procedures, discuss methods for pointwise estimation, and theoretically provide risk guarantees for density estimation under common assumptions.
- In experimental benchmarks, we find that properly tuned decoding-based regression heads are data-efficient and competitive with regular pointwise heads on tabular regression, yet are also expressive enough to perform against Gaussian mixture heads for density estimation.

## 2 Related Work and Motivation

The idea of *text-to-text regression* is especially relevant as LLMs are currently being fine-tuned as "Generative Reward Models" (Mahan et al., 2024; Zhang et al., 2024), i.e. end-to-end scoring methods for reinforcement learning feedback (Ziegler et al., 2019; Bai et al., 2022) or LLM-as-a-Judge (Zheng et al., 2023). Such reward modeling methods can be simpler than other forms such as Bradley-Terry (Bradley & Terry, 1952) which requires appending additional prediction heads and custom losses. However, little analysis has been done in isolation on the theoretical and modeling capabilities of using text, or more generally tokens, to represent real numbers. Understandably, one could argue that regular supervised fine-tuning over numbers represented as strings is unprincipled, considering that there is no notion of numeric distance when using cross-entropy loss.

However, we argue that token-based numeric modeling is actually natural, given observed phenomena and techniques proposed in recent works. Given a post-processed representation $\phi(x) \in \mathbb{R}^d$ after $x$ is sent through a task-specific encoder (MLP, CNN, etc.), we provide an overview of common regression heads $p_\theta(y|\phi(x))$ with trainable parameters $\theta$, which can be applied on top of $\phi(x)$ to return numeric outputs (additional techniques in Section 5).

**Pointwise Heads:** By far, the most commonly used regression head is a learnable deterministic function with weights $\theta$, typically a simple feed-forward network (often a single linear layer) mapping $\phi(x)$ to a single-point scalar, trained by minimizing a pointwise loss like mean squared error. To allow stability during training, the $y$-values must be normalized space, e.g. within $[0, 1]$.

**Parameteric Distribution Heads:** In the case of probabilistic outputs, one may apply distributions with parametric forms. A common example is a Gaussian head, e.g. $p_\theta(y|\phi(x)) = \mathcal{N}(\mu, \sigma^2)$ where $\mu, \sigma$ are deterministic learnable functions of $\phi(x)$. A more flexible variant is a finite mixture of Gaussians (Bishop, 1994), which can be extended to infinite mixtures (Rasmussen, 1999). Such mixture techniques can be more broadly seen within the realm of density estimation (Parzen, 1962; Rosenblatt, 1956) in which a complex distribution may be estimated using multiple simpler basis distributions.

**Histogram (Riemann) Distribution Heads:** One such basis common in deep learning applications is the piece-wise constant basis, for learning histograms over a finite support set $\{y_1, \ldots, y_n\} \subset \mathbb{R}$ via softmax parametrization, i.e. $p_\theta(y_i|\phi(x)) = \text{Softmax}^{(i)}(\phi(x)^T \cdot \theta)$ where $\theta \in \mathbb{R}^n$, which has led to strong results in value-based reinforcement learning (Imani & White, 2018; Bellemare et al., 2017) and tabular data (Hollmann et al., 2025; Chen et al., 2022). However, a drawback is that learning numeric distances between all of the bins requires more data as the size of the vocabulary increases. We term these as *Riemann* heads, following (Hollmann et al., 2025).

While there have been works on ordinal regression to learn rankings among these bins, such as using rank-consistency (Cao et al., 2020) and soft labels / metric-awareness (Diaz & Marathe, 2019), we propose a much simpler way, by simply considering the histogram distribution as a special case of decoding a sequence of length 1. By extending the sequence length instead, there can be an exponential reduction in bin count – e.g. $1000$ ($=10^3$) bins can be expressed instead using 10 bins and 3 decoding steps. While this intuitive idea has been studied in extreme classification problems (Wydmuch et al., 2018), it has not been thoroughly examined for numeric regression, which is the focus of our work. Below, we introduce decoding-based regression heads.

**Note:** To prevent confusion with the term "decoder" which is also a central component of generative models like Variational Autoencoders (VAEs) (Kingma & Welling, 2014), we stress a key distinction. While both VAE decoders and distributional regression heads map a feature vector to a probability distribution, their objectives differ: a VAE decoder models $p(x|\phi(x))$ to reconstruct the input $x$ from a latent $\phi(x)$, whereas a regression head models $p(y|\phi(x))$ to predict a separate target y. Given this difference in the output space, we avoid referring to standard distributional heads (e.g., Gaussian) as "decoders".

## 3 Decoding-Based Regression

In this work, we define the decoder head as an autoregressive sequence model, such as a compact Transformer decoder. Given a vocabulary $\mathcal{V}$, the Transformer takes the feature vector $\phi(x)$ as its initial context, and generates a discrete sequence of tokens $(t_1, \ldots, t_K) \in \mathcal{V}^K$ one token at a time. This sequence is a string representation of a number, and by modeling the probability $p_\theta(t_1, \ldots, t_K \mid \phi(x)) = \prod_{k=1}^{K} p_\theta(t_k \mid \phi(x), t_1 \ldots t_{k-1})$, the head implicitly defines a probability distribution over a discrete set of representable numbers. Below, we discuss natural token representations of numbers.

### 3.1 Numeric Token Representations

**Normalized Tokenization:** If $y$ is restricted to $[0, 1]$ (via scale normalization for example), then in Section 3.3 we show any smooth density $p(y|\phi(x))$ can be approximated with an increasing level of granularity as more tokens are used in the numeric representation, under some "universality" assumptions on $p_\theta$. This can be seen intuitively with a tree-based construction, i.e. for a base $B$, the vocabulary contains <0>, <1>, ..., <$B-1$>, and $y$ is simply represented by its base-$B$ expansion up to a length $K$. This setting aligns with standard data-science practices of also normalizing $y$-values according to training data or known bounds.

**Unnormalized Tokenization:** However, there are cases in which we would like to use an *unnormalized* tokenization scheme. Such cases include multi-task regression (Song et al., 2024), in which different tasks may have varying $y$-scales, or express very wide $y$-ranges for which appropriately normalizing $y$-values for the correct balance between numeric stability and expressiveness would be very tedious.

In this case, we may simply view normalized tokenizations as "mantissas" and then left-append sign and exponent tokens to form a base-$B$ generalization of the common IEEE-754 floating point representation (IEEE, 2019). Given length parameters $E$ and $M$, our tokenization is therefore <$s_e$><$e_1$>...<$e_E$><$m_1$>...<$m_M$> where $s_e, e_1, e_2, \ldots, e_E$ are the sign and base-B representation of the exponent and $m_1, m_2, \ldots, m_M$ are the most significant base-B digits of the mantissa. E.g. if ($B$=10, $E$=3, $M$=4), then $10^{-222} \times 1.23456789$ will be represented as <+><-><2><2><2><1><2><3><4>. Signs , <$s_e$> can have their own dedicated <->, <+> tokens or optionally reuse the <0>,<1> tokens from $\mathcal{V}$; this made little difference in results.

Note that the vocabulary can additionally contain "special" tokens for representing outputs not within a supported numeric range. For example, one can use a token <NaN> to represent non-numbers, commonly used in cases where $x$ may be an invalid input. We mention such cases for useful downstream applications, although the scope of this paper assumes $y$ is always within $\mathbb{R}$.

**Architecture:** Any autoregressive model can be used, so long as it supports constrained token decoding to enforce proper sequences which represent a valid number. By default, we use a small Transformer (Vaswani et al., 2017) due to its strong autoregression capabilities, with the initial token embedding as $\phi(x)$. As we show in our experiment section, this Transformer size can be made minimal, with negligible contribution to parameter count in comparison to the encoder.

Since the token space is finite while $\mathbb{R}$ is uncountable, this mapping is lossy (i.e. not invertible) and introduces a notion of *rounding error*. However, for large enough base $B$ and sequence lengths (both normalized and unnormalized), practically any $y$-value will be within the expressible range and rounding errors will be minimal. The trade-off is that the vocabulary size and sequential dependencies between tokens will also increase, and learning better numeric representations may thus require more training data. While it's possible to first pretrain these numeric representations as in Hollmann et al. (2025) for the histogram distribution, we show that with proper hyperparameter tuning, the Transformer decoder can be used out-of-the-box as a randomly initialized regression head.

## 3.2 Pointwise Estimation

In many cases, one may only be interested in a scalar quantity of interest $M(p_\theta)$ of the model's distribution (e.g. its mean). If $p_\theta$ matches the true distribution $p$ perfectly, then for a given pointwise loss $\ell : \mathbb{R}^2 \to \mathbb{R}$ the goal is then to select $M(p)$ which minimizes $\mathbb{E}_{y \sim p(\cdot|x)}[\ell(M(p), y)]$. It is well established that for common error functions like L2, L1, and L0, the optimal values are the mean, median, and mode of $p(\cdot|x)$, respectively. Lukasik et al. (2024) also leverage this observation to enhance LLM decoding.

To estimate these $M(p)$, the mode can be approximated using e.g. beam search (Graves, 2012), but efficiently estimating other common general pointwise representatives $M(p)$ from pure temperature samples is a broad topic - for example, one can efficiently approximate the true median from the Harrell-Davis estimator (Harrell & Davis, 1982), and more generally we refer the reader to Lehmann (1983) on statistical point estimators.

Especially for unnormalized tokenization, additional care needs to be taken, since in practice, the model can have a miniscule but non-zero probability of decoding an arbitrarily large outlier, even if the underlying true distribution is bounded. Such outliers can easily sway non-robust estimators such as the sample mean, as observed in Song et al. (2024). This issue fundamentally comes from the fact that some tokens are more significant than others, prompting the use of alternative tokenizations based on coding theory which are robust to corruptions, which we show can be effective in our experiment section.

Alternatively, decoding techniques from the LLM literature can also be used, e.g. top-$k$ (Fan et al., 2018) or top-$p$ (Holtzman et al., 2020), or even simply decreasing the temperature to increase model confidence and thereby filter out possible outliers. One can also avoid decoding altogether and use recently proposed RAFT (Lukasik et al., 2025; Chiang et al., 2025) which estimates $M(p)$ using a query-based approach using a finite fixed evaluation set $\mathcal{Y}$, e.g. for mean, $\mathbb{E}_{y \sim p_\theta}[y] \approx \frac{1}{N} \sum_{y' \in \mathcal{Y}} p_\theta(y') \cdot y'$, although the choice of $\mathcal{Y}$ may be non-trivial to obtain an unbiased estimate, especially over unnormalized tokenizations. This may also defeat the purpose of using a decoding head, which offers several density estimation benefits, as we discuss below. Overall, the choice of method for computing pointwise representations we leave as a hyperparameter to be tuned depending on the application.

## 3.3 Density Estimation and Theory

During training, to allow the full probabilistic modeling benefits of using a decoding head, we apply the standard cross-entropy loss over all sequence tokens. For a model $p_\theta$ and target $y = (t_1, \ldots, t_K)$, the cross-entropy loss (omitting $x$ to simplify notation) will be:

$$H(y, p_\theta) = \sum_{k=1}^{K} \sum_{\widehat{t}_k \in \mathcal{V}} -\mathbb{1}(\widehat{t}_k = t_k) \log p_\theta(\widehat{t}_k | t_1, \ldots, t_{k-1})$$

The expected loss over all $y$ sampled from the true distribution is then $\mathbb{E}_{y \sim p}[H(y, p_\theta)]$.

Given our tree-based tokenization and training loss, we provide formal guarantees for estimating one-dimensional densities on $[0, 1]$. Note that densities with finite support can be shifted and rescaled to have support in $[0, 1]$. Define $\lambda_k : [0, 1) \to \{0, 1\}^k$ be the operation that returns the first $k$ bits after the radix point in the (possibly infinite) binary representation of $y$. Concretely, if $y = 0.b_1 b_2 b_3 b_4\ldots$ then $\lambda_k(y) = (b_1, \ldots, b_k)$. We abuse notation and interpret $\lambda_k$'s output either as a sequence or as the real number it represents ($\sum_{i=1}^{k} b_i 2^{-i}$) depending on the context. The analysis is presented using binary (base-2) repre-

sentations (e.g. $\mathcal{V} = \{0,1\}$) for simplicity, but it holds for arbitrary bases. First, we provide an assumption on the learnability of our model and additional definitions:

**Definition 1** ($K$-bit universality). *Let $H(p,q) = \mathbb{E}_{y \sim p} - \log q(y)$ denote the cross-entropy between discrete distributions $p$ and $q$. Note that $H(p,p)$ is just the Shannon entropy of $p$. Call parametric model $p_\theta$ $K$-bit universal if for all discrete distributions $p$ on $K$-bit strings (equivalently, $2^K$ categories),*

$$\min_\theta H(p, p_\theta) = H(p, p).$$

*In other words, $p_\theta$ is $K$-bit universal if it is flexible enough to fit any discrete distribution on $2^K$ categories.*

**Definition 2.** *Define $p_\theta^k$ as probability of the first $k$ bits under $p_\theta$, marginalizing out the remaining bits. Concretely,*

$$p_\theta^k((b_1, \ldots, b_k)) = \sum_{\{b_{k+1}, \ldots, b_K\}} p_\theta((b_1, \ldots, b_K)).$$

*Seen another way, $p_\theta^k$ is the distribution over $k$-bit strings that results from auto-regressive decoding $p_\theta$ for exactly $k$ steps.*

**Definition 3.** *Let $f : [0,1] \to \mathbb{R}$ be a density function. With $\{Y_1, \ldots, Y_N\}$ as i.i.d. draws from $f$, define $\theta^*$ as the maximum likelihood estimator on the truncated sequence of $K$ bits. Concretely,*

$$\theta^*(Y_1, \ldots, Y_N) = \operatorname*{argmin}_\theta \frac{1}{N} \sum_{n=1}^N - \log p_\theta(\lambda_K(Y_n)).$$

*Define **risk** as the **mean integrated squared error** between true density $f$ and an estimator $\widehat{f}_N(Y_1, \ldots, Y_N)$:*

$$R(f, \widehat{f}_N) = \mathbb{E}_{Y_1, \ldots, Y_N \sim f} \int_0^1 \left( f(y) - \widehat{f}_N(y) \right)^2 dy.$$

We now give our main result below, expressing the distributional fit in terms of bias and variance. The proof is deferred to Appendix B.

**Theorem 1.** *Assume our decoding-based regression model $p_\theta : \{0,1\}^K \to \Delta^{2^K}$ is $K$-bit universal, and $f$ be any twice continuously differentiable density function. If the maximum likelihood estimator at $k$ is $f_N^{k*}(y) = 2^k p_{\theta^*(Y_1, \ldots, Y_N)}^k(\lambda_k(y))$ for $y \in [0,1]$, then the risk can be exactly computed:*

$$R\left(f, f_N^{k*}(y)\right) = \underbrace{\frac{2^{-2k}}{12} \int_0^1 f'(y)^2 dy}_{Bias} + \underbrace{\frac{2^k}{N}}_{Variance} + \underbrace{O(2^{-4k} + 1/N)}_{Negligible}, \quad \forall k \leq K.$$

Note that this theorem is broad, applicable to both Riemann and decoding heads even if they perform inference at a lower token length $k$ than the maximum length $K$ used during training. For simplicity, let us assume that the maximal length is always used (i.e. $k = K$). Intuitively, this implies that one needs a higher resolution $K$ to capture the curvature of $f$, but as the number of bins increases, more data points $N$ are required to learn to separate these $2^K$ bins. In Figure 2, for large $N=16384$, we show this trend holds empirically where there is an optimal $K \approx 5$ which minimizes the error.

When $N$ is quite small (e.g. 1024) we see that the decoder head significantly deviates from the theoretical risk (for the better) when the number of bits is large ($>9$), while the Riemann head still fits it tightly. Recall that we required a "universality" assumption, which says that our model can learn any discrete distribution over $K$-bit strings perfectly. We can decompose this assumption further into two pieces: 1) that there exists $\theta^*$ in our model class that achieves the minimum cross-entropy (i.e. $p_{\theta^*} = p$ in Definition 1), and 2) that our SGD-based training procedure is able to find it. An explanation for this phenomenon is that in this regime (low sample size and large number of bits, or equivalently, a large number of bins), the risk profile of

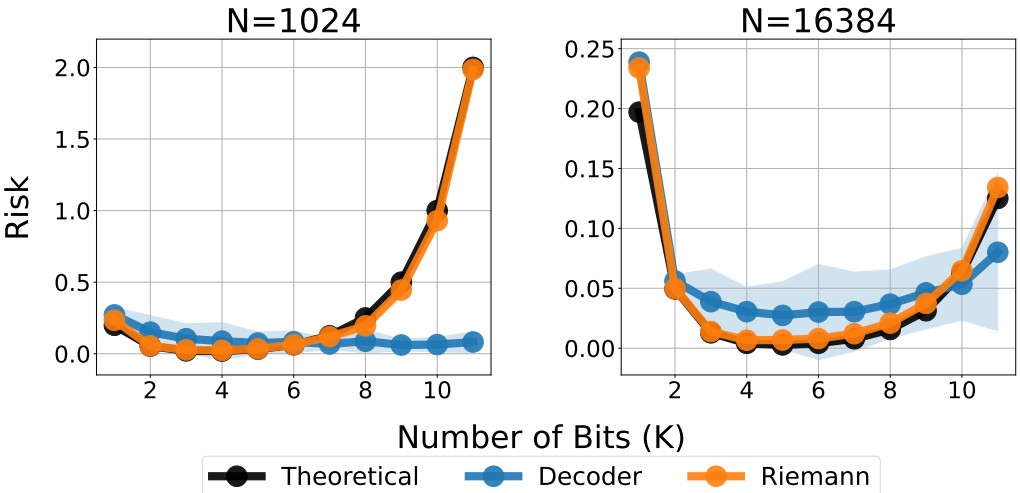

Figure 2: Lower ($\downarrow$) is better. Risk (theoretical and empirical) when varying $K$ and $N$ to fit a truncated $\mathcal{N}_{[0,1]}(0.5, 0.25^2)$ distribution using binary tokenization. Results averaged across 10 runs each.

the classical Riemann estimator is dominated by the variance term. Few samples land in each bin and as a result the histogram-based density estimate for the bins is noisy.

It is conceivable that a combination of the inductive bias of our model class and the implicit bias of our SGD training procedure makes the decoder less likely to fit noise; a concrete example would be that the model is biased to learn smooth distributions, and so when asked to fit the highly discontinuous empirical distribution arising from dropping few samples into a large number of bins, it refuses to, instead opting to learn a smooth approximation, and thereby driving down the variance term and hence the overall risk. This suggests the decoder head possesses implicit regularization properties which make it much more efficient with low training data.

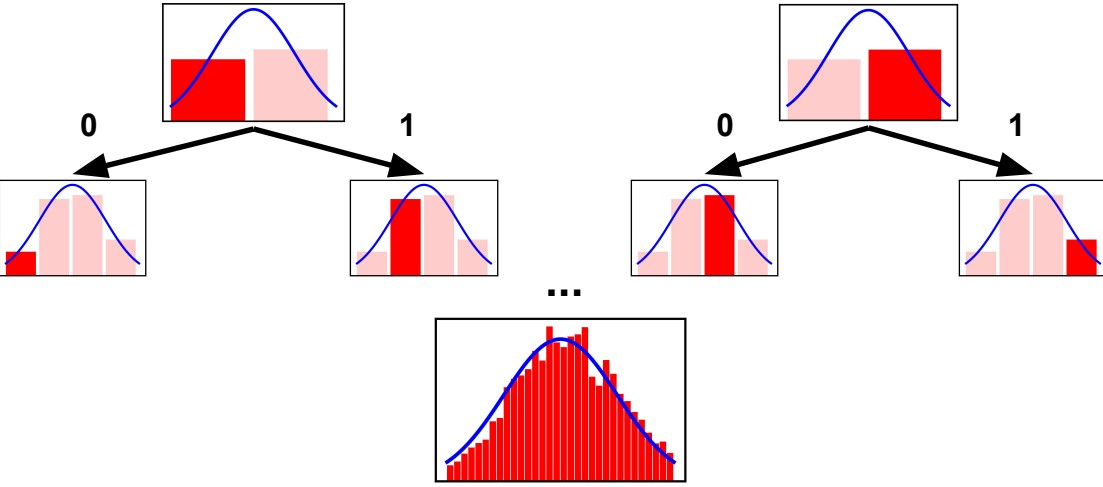

Figure 3: Visualization of fitting a truncated Gaussian distribution. Each level $k$ of the binary tree represents the empirical fit using $k$ bits, and each bin gets subdivided into two.

Taking a closer look at the decoding mechanism, a crucial observation is that $\lambda_k$ essentially discretizes the unit interval (and hence $f$ as well) into bins $\{B_j\}_{j=0}^{2^k-1}$, where $B_j = [j2^{-k}, (j+1)2^{-k})$ so that $\mathbb{P}(x \in B_j) = \int_{B_j} f(y)dy$. We can identify $k$-bit sequence $y = 0.b_1 \ldots b_k$ with the interval $[y, y + 2^{-k}]$. With a single bit ($K = 1$) we learn a histogram estimator on two bins $[0, 1/2)$ and $[1/2, 1)$ representing 0 and 1. With

two bits we refine our prediction using four bins: $[0, 1/4), [1/4, 1/2), [1/2, 3/4),$ and $[3/4, 1)$ representing $(0, 0), (0, 1), (1, 0), (1, 1)$ respectively (because, for example $(0, 1)$ means $0.01_2 = 1/4$).

We can interpret binary representations in terms of binary trees on $2^K$ leaf nodes where nodes represent intervals (the root representing $[0, 1)$) and left and right children represent the left and right halves of the node's interval. Reading off bits tells us how to traverse this tree, where 0 and 1 mean traverse the left and right subtrees respectively. For example, to arrive at $(0, 1, 1) = 0.011_2 = 3/8$ our traversal is: $[0, 1) \rightarrow [0, 1/2) \rightarrow [1/4, 1/2) \rightarrow [3/8, 1/2)$.

When trained on $K$-bit sequences, our decoding head $p_\theta$ *simultaneously* learns $K$ histogram estimators for $f$; $2^k p_\theta^k(\lambda_k(y))$ is the $k$-th histogram estimator (over $2^k$ bins). In other words, as we decode bits one-by-one auto-regressively, we are iteratively refining our prediction. Figure 3 shows this mechanism in detail in the case of fitting a truncated Gaussian distribution.

There are alternatives to binary representations, for example *p-adic expansions*, or even the *Stern–Brocot tree* which uses the median to determine the child-parent relationship. An interesting research question left for future work is whether these more exotic representations of real numbers are better suited for our sequence-based regression model than the standard representations.

## 4 Experiments

Our main goals for experiments are to:

- Demonstrate decoder heads can be effective swap-in replacements to common pointwise regression heads.
- Establish the density estimation capabilities of the decoding-based head over any distribution over $\mathbb{R}$.
- Ablate the effect of decoder head size and sequence-specific methods such as error correction on performance.

To maintain fairness, all neural network methods have access to the same encoder $\phi(x)$, which is a large multi-layer perceptron (MLP) with ReLU activations, with hyperparameter sweeps over number of layers (up to 5) and hidden unit sizes (up to 2048). Furthermore, the decoder head uses only 1 layer and 32 units, making up for less than 10% of the total network parameter count, which minimizes its contribution to representation learning as a confounding factor.

Furthermore, for distributional heads (e.g. decoder, Riemann), we sweep their specific settings (e.g. number of bins / tokenization) over reasonable values - additional details are found in Appendix C. For the vast majority of tabular regression problems, we found that the process of training and tuning only requires at most 20 minutes on a single Nvidia P100 GPU, making the decoder head relatively cheap to use. For comparisons, we use relative mean squared error within individual tasks and scale-invariant Kendall-Tau correlation for aggregate comparisons.

### 4.1 Curve Fitting

We begin by visually demonstrating the fundamental representation power of tokenization. In Figure 4, the unnormalized decoder head is able to successfully capture the shapes of various functions with which both the Riemann and pointwise head struggle. The issue with using the pointwise head stems from two main factors: (1) requiring $y$-normalization, which leads to numeric instabilities especially with functions with very high or unbounded $y$-ranges, and (2) struggling to model abrupt or high rates of change (i.e. large Lipschitz

| Regression Head | Input Dimension | | | |
|---|---|---|---|---|
| | 5 | 10 | 15 | 20 |
| Unnormalized Decoder | 89.56 | 88.71 | 87.49 | 86.11 |
| Normalized Decoder | 89.40 | 88.54 | 86.90 | 86.02 |
| Pointwise | 89.08 | 88.25 | 88.06 | 86.78 |
| Riemann | 88.94 | 88.30 | 87.42 | 86.78 |

Table 1: Higher (↑) is better. Mean Kendall-Tau correlations over BBOB functions with ($\approx$100K) training data. Individual function results can be seen in Appendix A.2.

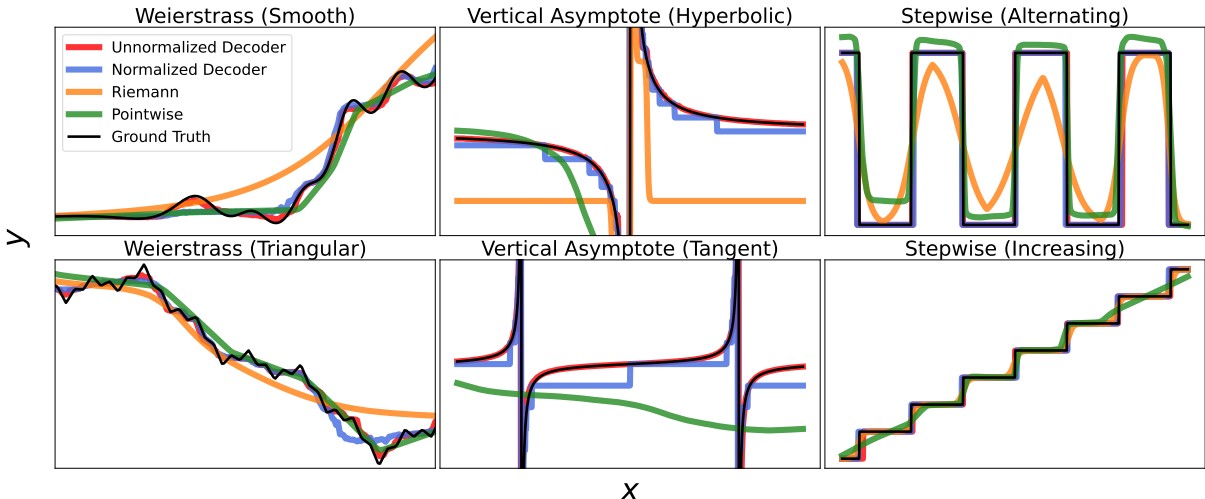

Figure 4: Visual fit to ground truth is better. Curve fitting plots for various 1D functions. Both models are trained over 100K $(x, y)$ points, where $x$ is uniformly sampled from a bounded range. Note: These results occur regardless of xy-scales, which are omitted for brevity. Riemann prediction for "Vertical Asymptote (Tangent)" went out of range.

constants). In contrast, the unnormalized decoder head does not encounter these issues due to its ability to express a very high range of $y$-values, with the normalized decoder also performing decently.

In Table 1, as a sanity check over higher-dimensional functions, synthetic continuous objectives from the Black-box Optimization Benchmarking (BBOB) suite (Elhara et al., 2019) can also be sufficiently fitted by both the unnormalized and normalized decoder heads just as well as the pointwise and Riemann heads.

## 4.2 Real-World Regression

In Figure 5, over real-world OpenML (Vanschoren et al., 2013) regression tasks from OpenML-CTR23 (Fischer et al., 2023) and AMLB (Gijsbers et al., 2024), using the unnormalized decoder head is competitive to using a regular pointwise head given the same training data. In fact, in the majority of tasks, the decoder outperforms the pointwise head, and in a few cases, the gap can be quite significant ($>0.3$). Full results in Appendix A.3, Figure 13 also lead to the same conclusion for normalized decoder heads.

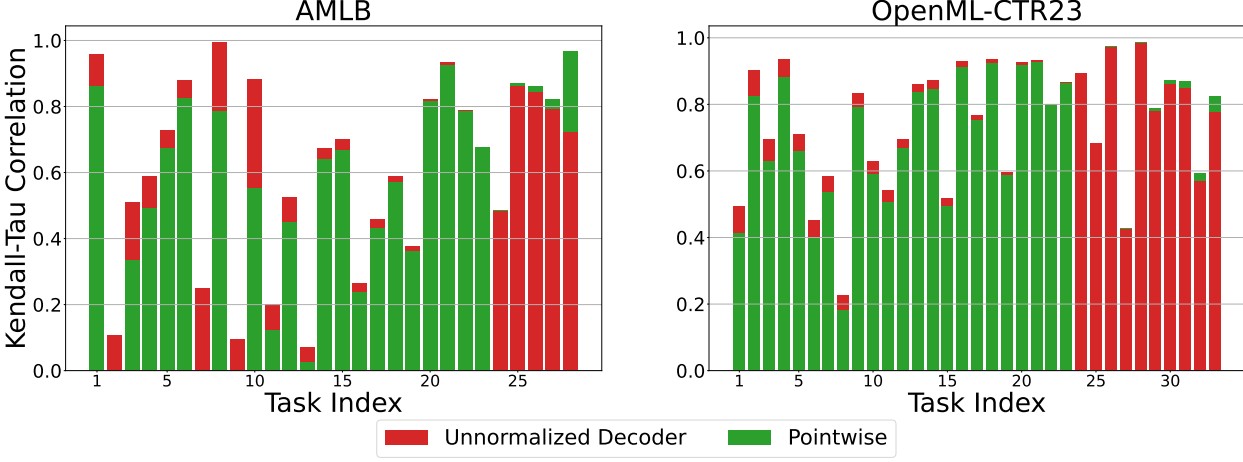

Figure 5: Higher (↑) is better. Kendall-Tau regression scores over AMLB and OpenML-CTR23 tasks using up to 10K maximum training points. Each bar averaged over 10 runs and bars from the same task (but different regressors) are stacked on top of each other and sorted by gap performance gap.

In Figure 6, both decoding heads outperform the Riemann head in the vast majority of tasks as well, suggesting improved sample efficiency from minimizing vocabulary / bin sizes.

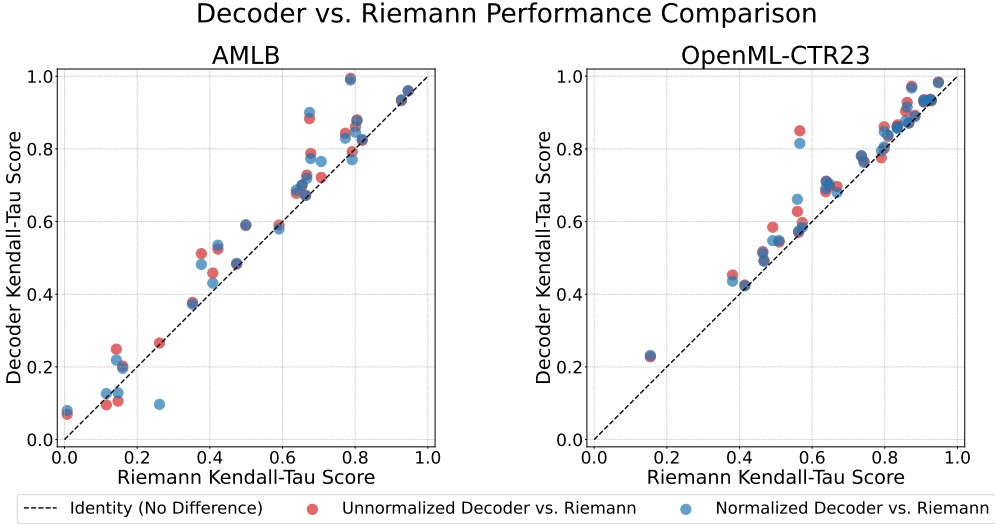

Figure 6: Upper left (↖) is better for decoder heads. Paired scatter plots for comparing both decoders against the Riemann head, over real-world regression tasks.

In order to more rigorously validate the sample efficiency hypothesis, in Figure 7, we compare the use of all normalized heads (normalized decoder, Riemann histogram, and pointwise), when varying the amount of training data. We omit unnormalized decoder results, as it aggregates samples differently. We first observe the data inefficiency of using the histogram head on selected regression tasks - in certain cases, the histogram head plateaus, unable to even achieve the performance of the decoder head, regardless of the amount of training data.

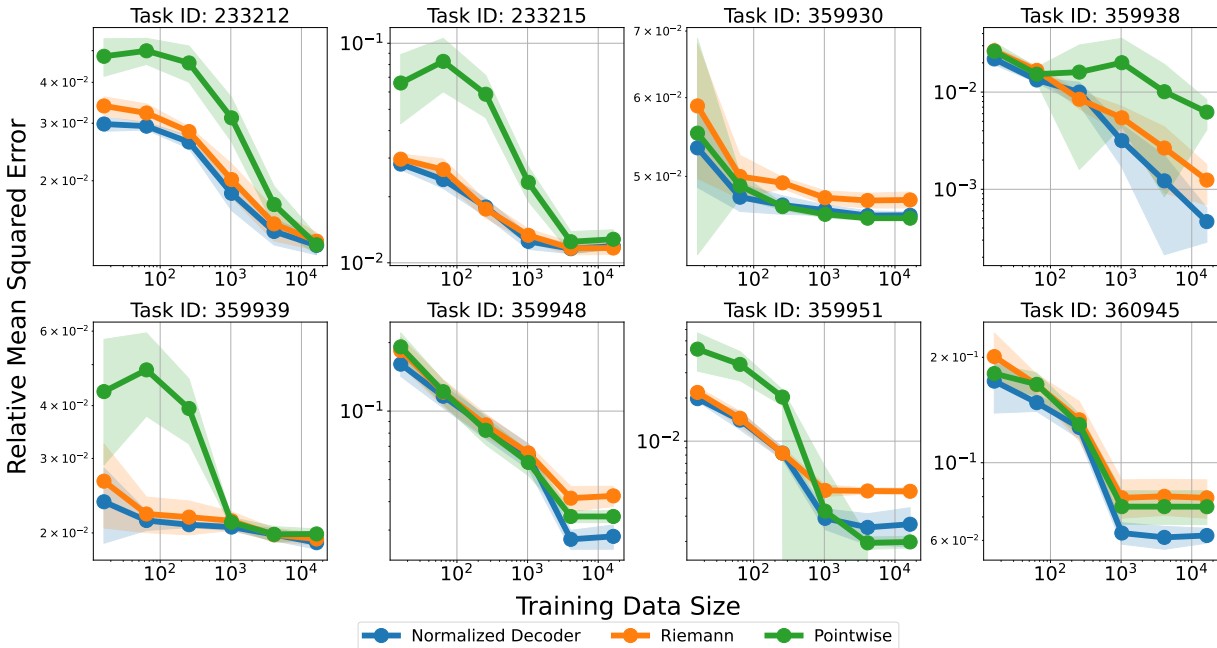

Figure 7: Lower (↓) is better. Relative mean squared error (MSE) over selected AMLB tasks. Each method used a min-max linear scaling normalization on $y$-values. Full results in Appendix A.1, Figure 11.

Furthermore, interesting observations can be made when comparing against the standard pointwise head as a baseline. In high data regimes ($\approx 10^4$ data points), there are cases in which it *also* plateaus earlier than the decoding head. In low data regimes ($\approx 10^1$ data points), one would expect the decoding head to struggle more as it needs to learn numeric token representations, but as it turns out, the pointwise head can perform worse due to numeric instabilities of its own. Due to undertraining, the pointwise head required appending a sigmoid activation to enforce the normalized output to be within [0,1] to avoid extremely high MSE errors.

## 4.3 Density Estimation

In Figure 8, we further see the decoding head's ability to perform density estimation over various shapes. Given unbounded training data it is able to capture the overall distribution $p(y|x)$ well, although there can be slight outlier noise as shown by lighter points. In Appendix A.6 we show that even baseline heads such as Mixture Density Networks (MDNs) (Bishop, 1994) and Riemann distributions also suffer from noisy outputs. While one can enforce the sampling to be tighter (e.g. lowering temperature) to remove noise, this tighter sampling can unfortunately also reduce expressivity. In general, we find that vanilla temperature sampling with temperature $\approx 1.0$ is the best way to match $p(y|x)$.

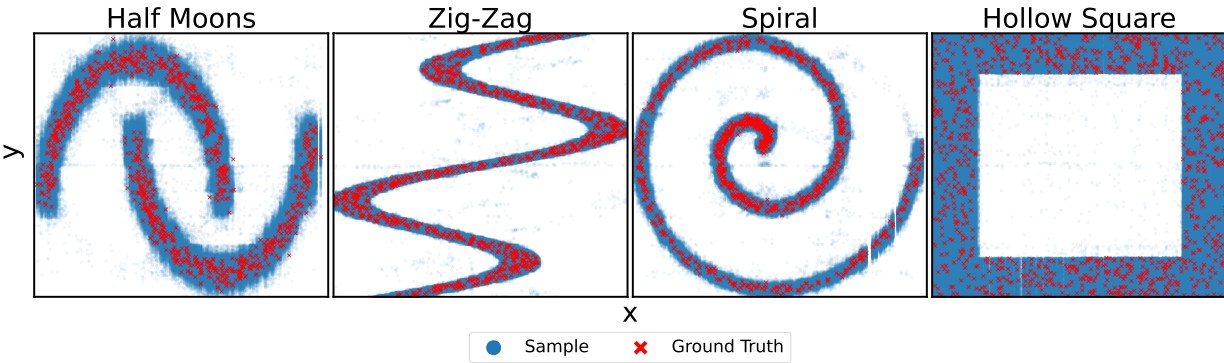

Figure 8: Fit to ground truth is better. Density estimation visualization over various shapes using an unnormalized decoder head with vanilla temperature sampling. Note that these results occur regardless of xy-scales, which are omitted for brevity.

In Table 2, we display the negative log-likelihood (NLL) on a collection of representative real-world datasets from the UCI regression repository (Dua & Graff, 2017) (full results over 25 datasets in Appendix A.5). We see that MDN head performance has high variability, at times able to perform the best but also extremely poorly depending on the task. Meanwhile both decoding heads remain reliable overall (NLL<0.7 always). In comparison, the Riemann head consistently underperforms in every task.

| Dataset | MDN | UD | ND | R |
|---|---|---|---|---|
| Airfoil | **0.12 ± 0.11** | 0.40 ± 0.01 | 0.34 ± 0.01 | 1.33 ± 0.14 |
| Bike | 4.59 ± 0.86 | 0.12 ± 0.00 | **0.10 ± 0.01** | 0.36 ± 0.05 |
| Elevators | 0.30 ± 0.43 | 0.15 ± 0.00 | **0.13 ± 0.00** | 1.12 ± 0.02 |
| Gas | 0.68 ± 0.25 | **0.02 ± 0.01** | **0.02 ± 0.00** | 0.20 ± 0.09 |
| Housing | **0.22 ± 0.13** | 0.41 ± 0.03 | 0.38 ± 0.03 | 1.56 ± 0.21 |
| Kin 40K | 7.49 ± 0.73 | 0.19 ± 0.01 | **0.12 ± 0.01** | 0.39 ± 0.03 |
| Pol | 1.49 ± 0.41 | **0.01 ± 0.00** | **0.01 ± 0.00** | 0.18 ± 0.02 |
| Protein | 1.07 ± 0.44 | **0.34 ± 0.00** | 0.41 ± 0.01 | 1.55 ± 0.04 |
| Pumadyn32nm | 0.69 ± 1.26 | **0.55 ± 0.00** | 0.58 ± 0.02 | 2.32 ± 0.03 |
| Wine | **0.05 ± 0.12** | 0.24 ± 0.01 | 0.21 ± 0.01 | 1.67 ± 0.14 |
| Yacht | **0.21 ± 0.10** | 0.39 ± 0.02 | 0.23 ± 0.05 | 1.29 ± 0.38 |

Table 2: Lower (↓) is better. Avg. NLL (± StdDev) of test examples on UCI datasets over 10 train-test splits. Abbreviations: (UD, ND) = (unnormalized, normalized) decoder heads respectively; R = Riemann.

## 4.4 Ablation: Role of Decoding Head Size

We ablate the effect of the decoding head's size on performance. We first fix the tokenization for the normalized decoding head ($B$=10, $K$=4) and then sweep the number of layers, heads, and hidden units. In Figure 9, we observe that larger decoding heads do sometimes help, but only up to a certain point, at which overfitting can occur. This was also observed over regression over BBOB functions and with the unnormalized decoding head, but we omitted these results for brevity.

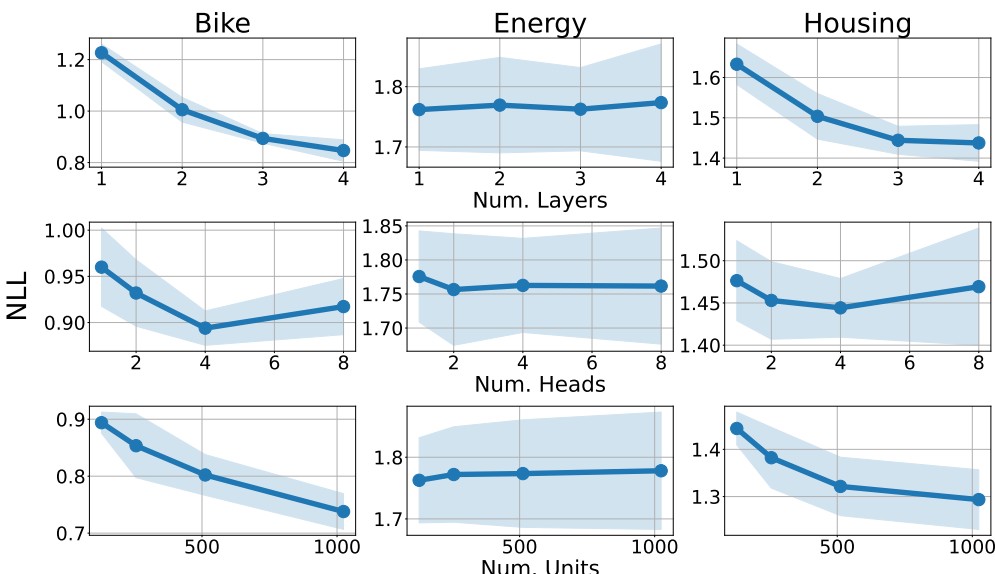

Figure 9: Lower ($\downarrow$) is better. NLL over UCI datasets, when varying different axis (layers, heads, units) from a fixed default of (3, 4, 128) respectively.

## 4.5 Ablation: Error Correction

One can also improve regression behavior using techniques purely by modifying sequence representations. Inspired by the field of coding theory, we can use *error correction*, where we may simply have the decoding head repeat its output multiple times $(t_1, \ldots, t_K, t'_1, \ldots, t'_K, t''_1, \ldots, t''_K, \ldots)$ during training, and at inference perform majority voting on each location $k \in \{1, \ldots, K\}$.

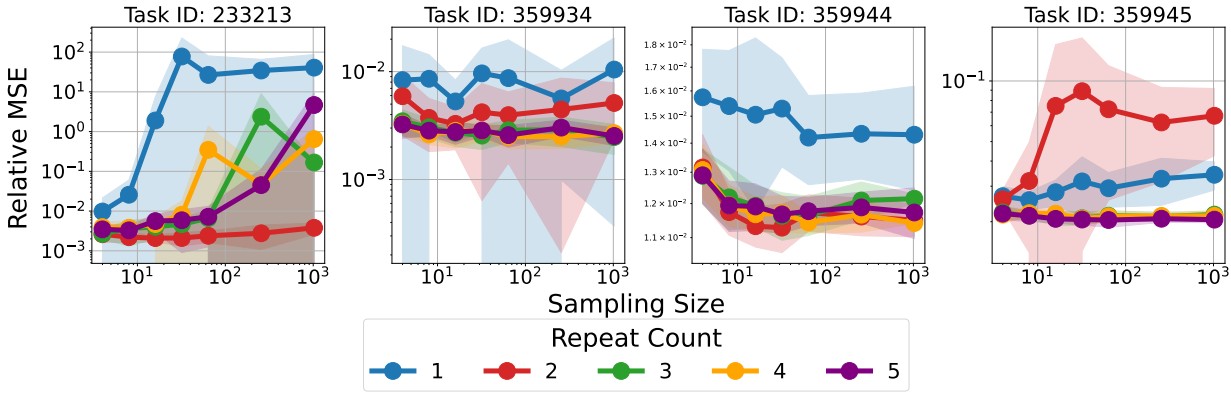

Figure 10: Lower ($\downarrow$) is better. Relative MSE over selected AMLB tasks, when varying output repetitions.

In Figure 10, we focus on the unnormalized case when using mean aggregation, where performance can be significantly harmed from extreme outliers. We see that when using regular tokenization (repeat count=1),

as more samples are drawn, the likelihood of drawing outliers increases the error. However, the error can be substantially decreased by training the decoding head to decode the same tokens repeatedly and allow better scaling with samples, although repeating too many times may make learning more difficult. Not all error correction techniques improve results however - in Appendix A.4, we briefly observe negative results applying other types of error correction, and we leave exploring the space of such methods for future work.

## 5   Discussion: Limitations and Extensions

This work establishes the validity of training decoding heads with cross-entropy losses for regression. To minimize confounding factors, our designs remained very simple (e.g. using basic primitives such as softmax and vanilla attention mechanisms) yet principled (e.g. digit-by-digit tokenization and constrained sampling). We list some limitations of this work, along with more potential areas for exploration.

**Modern LLM Architectures:** Many modern LLM architectures no longer use vanilla attention mechanisms, instead opting for sparsity or low-rank approximations. In addition, MLPs have typically been replaced by mixtures of experts for memory reduction. It is worth studying further how these changes affect the performance of numeric decoding.

**Tokenization and Sampling:** Our digit-by-digit tokenization and constrained decoding can be considered ideal cases, which modern LLMs only approximately implement. The main sources of differences include vocabularies which do not use individual digit tokens, and sampling procedures which are Top-P or Top-K, which may accidentally choose invalid tokens. We hypothesize that these issues can decrease regression performance.

**Intermediate Language and Training:** In many LLM use-cases involving numeric prediction problems, the decoder not only must return a tokenized number, but also other intermediate language tokens. Our work does not address these cases, and it is unclear how fine-tuning on these intermediate language tokens may affect training dynamics and numeric prediction performance. Furthermore, modern LLM training consists of reinforcement learning (RL), and it remains to be studied how reward-based training methods affect regression results.

**Multi-objective Regression:** In this paper, we only studied the single-objective regression case. However, one may easily modify our paradigm to support multiple objectives $p(y^{(1)}, \ldots, y^{(M)} \,|\, \phi(x))$, e.g. by decoding a concatenated sequence of those objectives. This has the particular benefit of modeling objectives autoregressively, which can be difficult to perform with classical techniques. The benefit of multi-objective density estimation is even more pronounced, due to the decoder's universal approximation abilities.

**Other Regression Architectures:** We did not compare to significantly more complex methods such as stochastic networks, which use stochastic activations or weights. Early examples include Sigmoid Belief Nets (Neal, 1992; Tang & Salakhutdinov, 2013), which have not seen wide adoption due to their complex architectures and expectation-maximization training updates. Bayesian neural networks (Lampinen & Vehtari, 2001; Titterington, 2004; Goan & Fookes, 2020) can be seen as more modern stochastic networks, but still possess complex inference techniques, e.g. Markov Chain Monte Carlo (MCMC) or variational inference. Similarly, Energy-based models (Teh et al., 2003) can be used for image-based regression (Gustafsson et al., 2020; Liu et al., 2022) but still see limited use due MCMC required at inference time.

## 6   Conclusion and Future Work

We thoroughly investigated the many benefits but also drawbacks of using decoding-based regression. We described a natural tokenization scheme for both normalized and unnormalized $y$-values, and theoretically established its risk minimization properties. Empirically, we showed that it can be competitive as, or even outperform traditional pointwise heads for regression tasks. Furthermore, it is also capable of density estimation over a variety of conditional distributions $p(y|\phi(x))$, and can further outperform common baseline regression heads such as Gaussian mixtures and histogram distributions. We hope this work will also be a valuable reference for the language modeling community and that it provides a principled explanation for the use of supervised fine-tuning over numeric targets.

## Acknowledgements

We would like to thank Yutian Chen for his valuable review of the work and Bangding (Jeffrey) Yang for technical help. We further thank Yash Akhauri, Aviral Kumar, Bryan Lewandowski, Michal Lukasik, Sagi Perel, David Smalling, and Subhashini Venugopalan for useful discussions, and Daniel Golovin and Denny Zhou for continuing support.

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

# Appendix

# A  Additional Experiments

## A.1  Data Scaling: Extended

For completeness, we display the plots over all tasks in AMLB (Gijsbers et al., 2024). We confirm the data-efficiency of the decoder head against the Riemann distribution head on nearly every regression task. Furthermore, we observe numerous cases where both distributional methods outperform the pointwise head, especially in low data regimes.

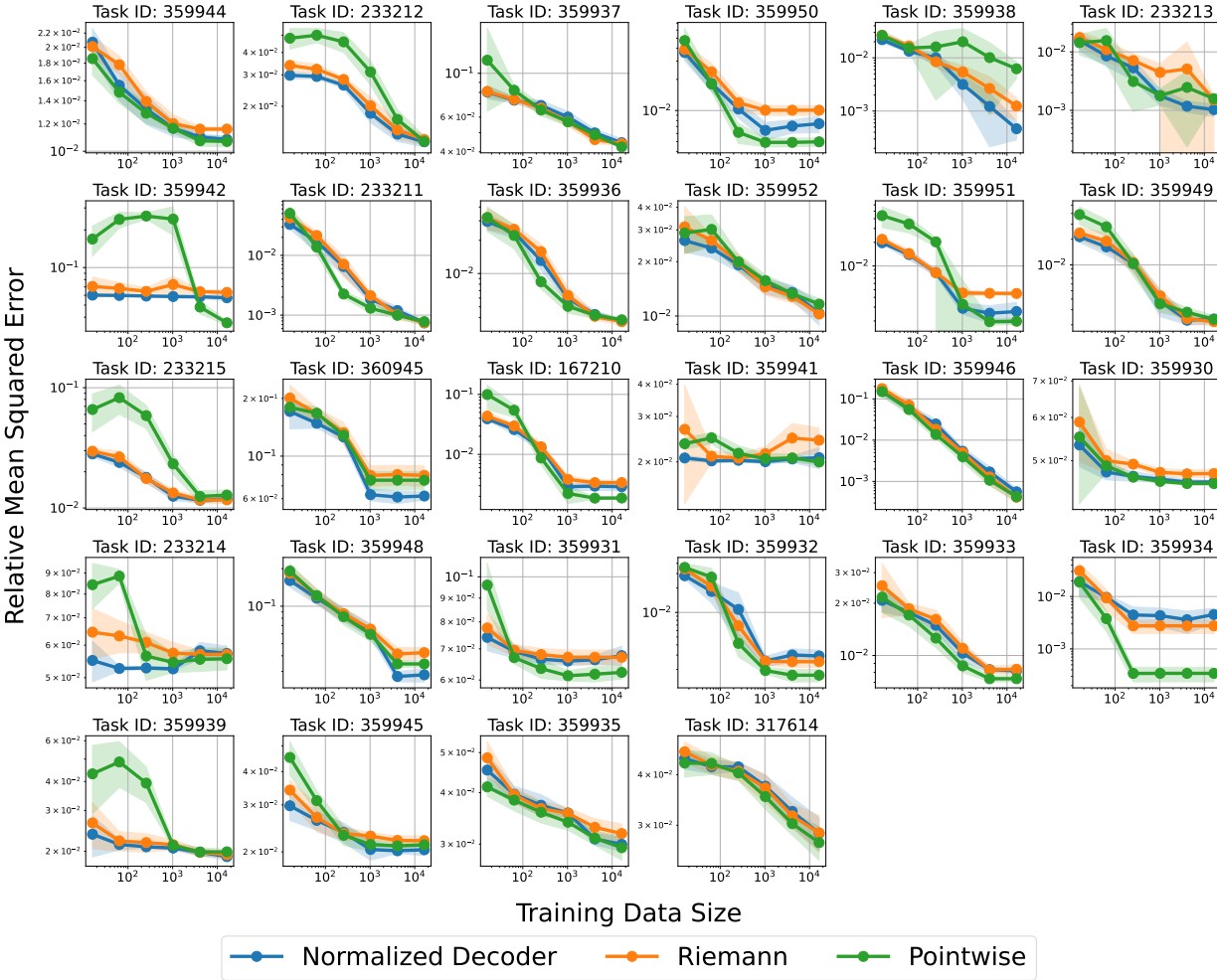

Figure 11: Lower (↓) is better. Regression performance as a function of training data scaling between using the normalized decoder vs. Reimannian distribution as regression heads. Each point was averaged over 10 training runs over random combinations of datapoints from the original AMLB task's training set.

## A.2 BBOB Curve Fitting: Extended

In Figure 12, we compare the curve fitting properties of multiple regression heads. We see overall that the decoder head is competitive and has both pros and cons for specific function landscapes from the BBOB benchmark.

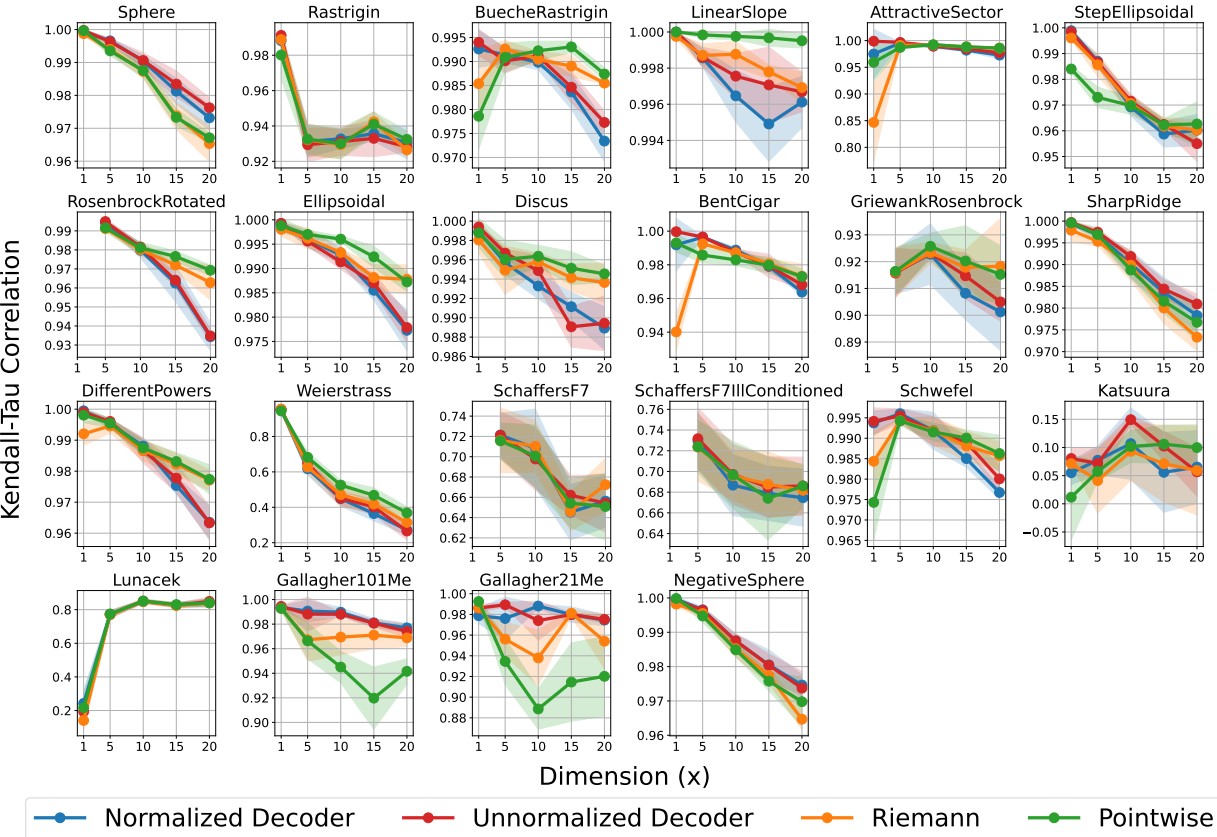

Figure 12: Higher ($\uparrow$) is better. Extended results from Table 1 in the main body. Regression performance as a function of input dimension over BBOB functions using Kendall-Tau correlation. Each point was averaged over 10 training runs, each with 100K training points $(x, y)$ where each $x$ is sampled uniformly from $[-5, 5]$ coordinate-wise. **Note:** Some functions such as RosenbrockRotated or GriewankRosenbrock are undefined when dimension is 1, so we skip those points.

### A.3 Individual OpenML Kendall-Taus

In Figure 13, we present full results on the AMLB and OpenML-CTR23 regression benchmarks, over all regression heads. We see that both the unnormalized and normalized decoder heads remain competitive throughout the benchmarks.

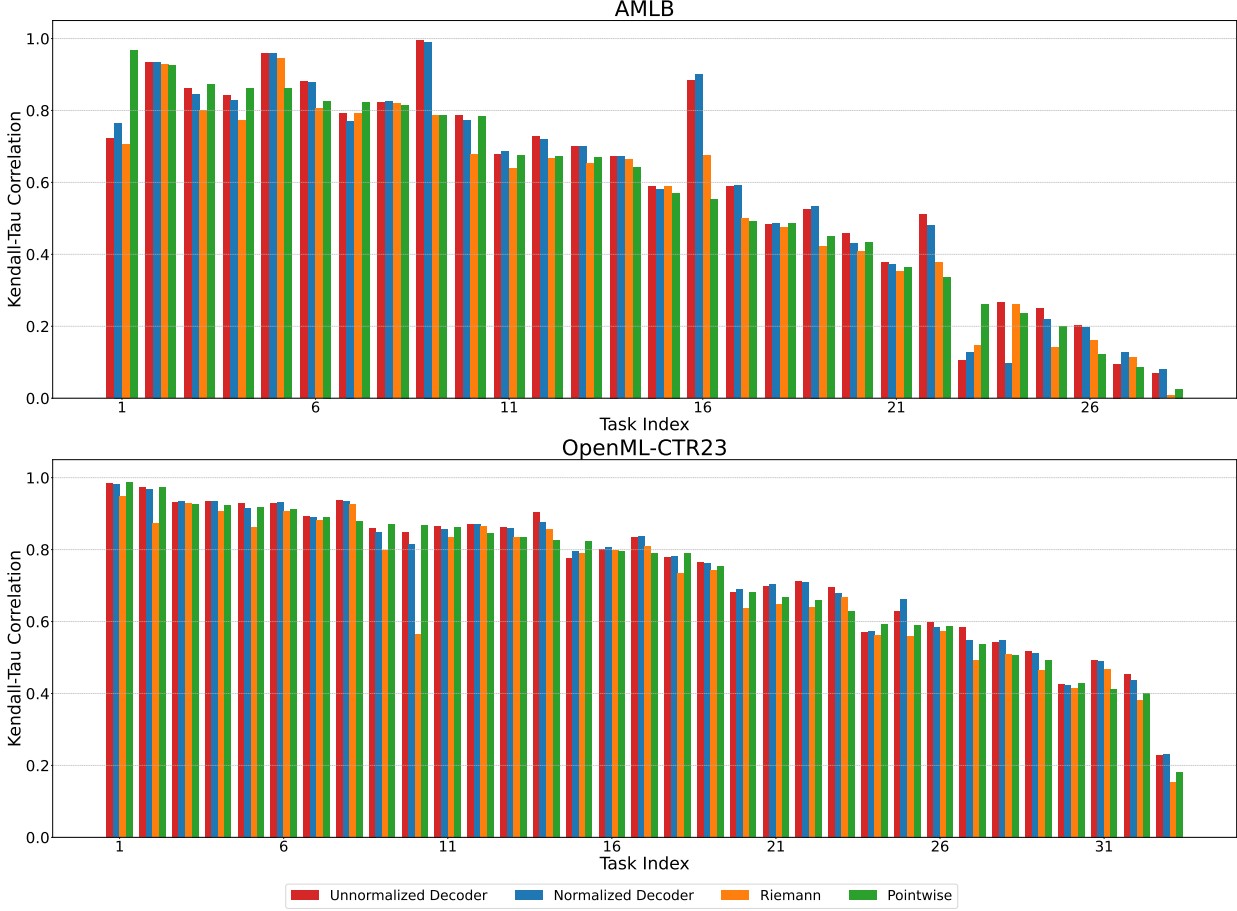

Figure 13: Higher (↑) is better. Extended results from Figure 5, but for all four regression heads on all tasks. Task IDs sorted by pointwise head performance.

### A.4 Alternative Tokenization Schemes: Hamming-Distance

One possible criticism of the default tree-based tokenization in the normalized decoding case, is the vulnerability to small changes in the left-most significant tokens, which can cause large numeric changes in the actual number. Qin (2018) notes this and proposes an alternative "Hamming Distance-based" binary representation which is robust to bitwise edits, and upper bounds the possible distortion $|y' - y|$ as a function of the edit distance between the Hamming representations of $y'$ and $y$. For example, if the binary length is 3, the representation for all integers $\{0, 1, \ldots, 2^3\}$ is $\{(000), (001), (010), (100), (011), (101), (110), (111)\}$ which can also be used in the normalized case for $\{0/2^3, 1/2^3, \ldots, 7/2^3\} \subset [0, 1]$. In Figure 14, we show however, such a representation may *not* lead to better regression results, which we hypothesize is due to this representation being more difficult to learn.

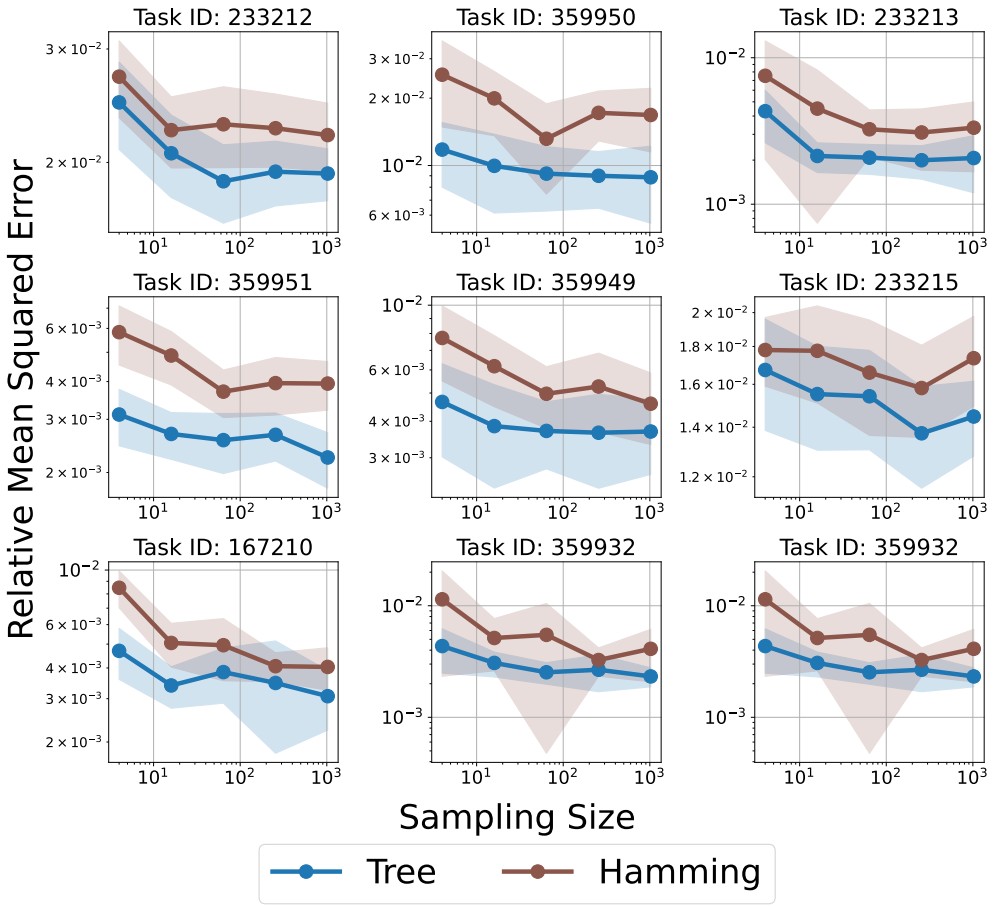

Figure 14: Lower ($\downarrow$) is better. Regression performance while vary sampling size from $y \sim p_\theta(\cdot|x)$ using binary tree-based tokenization vs. Hamming representation on normalized decoder with mean aggregation. Each point was averaged over 10 training runs over random size-1000 combinations of the original AMLB task's training data points.

## A.5 Full UCI Density Estimation Results

We obtained the datasets from `https://github.com/treforevans/uci_datasets` which preprocessed and scraped the data from the official UCI website at `https://archive.ics.uci.edu/`.

| Dataset | Mixture Density Network | Unnormalized Decoder | Normalized Decoder | Riemann |
|---|---|---|---|---|
| Airfoil | **0.12 ± 0.11** | 0.40 ± 0.01 | 0.34 ± 0.01 | 1.33 ± 0.14 |
| AutoMPG | **0.21 ± 0.07** | 0.32 ± 0.03 | 0.41 ± 0.05 | 1.62 ± 0.17 |
| Autos | **0.32 ± 0.23** | 0.48 ± 0.05 | 0.47 ± 0.07 | 2.60 ± 0.76 |
| Bike | 4.59 ± 0.86 | 0.12 ± 0.00 | **0.10 ± 0.01** | 0.36 ± 0.05 |
| BreastCancer | **0.32 ± 0.09** | 0.48 ± 0.05 | 0.64 ± 0.03 | 2.85 ± 0.37 |
| Challenger | **-0.29 ± 0.66** | 0.14 ± 0.14 | 0.06 ± 0.08 | 0.87 ± 0.77 |
| Concrete | **0.15 ± 0.05** | 0.43 ± 0.03 | 0.41 ± 0.04 | 1.67 ± 0.20 |
| Elevators | 0.30 ± 0.43 | 0.15 ± 0.00 | **0.13 ± 0.00** | 1.12 ± 0.02 |
| Energy | 0.40 ± 0.14 | 0.17 ± 0.03 | **0.16 ± 0.05** | 0.38 ± 0.20 |
| Fertility | **-0.06 ± 0.16** | 0.31 ± 0.09 | 0.46 ± 0.13 | 2.41 ± 0.61 |
| Gas | 0.68 ± 0.25 | **0.02 ± 0.01** | **0.02 ± 0.00** | 0.20 ± 0.09 |
| Housing | **0.22 ± 0.13** | 0.41 ± 0.03 | 0.38 ± 0.03 | 1.56 ± 0.21 |
| KeggDirected | 2.41 ± 1.10 | **0.05 ± 0.00** | **0.05 ± 0.00** | 0.22 ± 0.02 |
| Kin 40K | 7.49 ± 0.73 | 0.19 ± 0.01 | **0.12 ± 0.01** | 0.39 ± 0.03 |
| Parkinsons | 0.59 ± 0.18 | 0.40 ± 0.02 | **0.39 ± 0.03** | 1.40 ± 0.33 |
| Pol | 1.49 ± 0.41 | **0.01 ± 0.00** | **0.01 ± 0.00** | 0.18 ± 0.02 |
| Protein | 1.07 ± 0.44 | **0.34 ± 0.00** | 0.41 ± 0.01 | 1.55 ± 0.04 |
| Pumadyn32nm | 0.69 ± 1.26 | **0.55 ± 0.00** | 0.58 ± 0.02 | 2.32 ± 0.03 |
| Slice | 7.09 ± 0.09 | 0.05 ± 0.00 | **0.02 ± 0.00** | 0.08 ± 0.02 |
| SML | 1.31 ± 0.59 | 0.21 ± 0.01 | **0.11 ± 0.01** | 0.35 ± 0.03 |
| Solar | **-1.40 ± 0.29** | 0.04 ± 0.01 | 0.04 ± 0.01 | 0.61 ± 0.12 |
| Stock | **-0.15 ± 0.15** | 0.27 ± 0.04 | 0.32 ± 0.04 | 1.63 ± 0.46 |
| TamiElectric | **0.01 ± 0.00** | 0.46 ± 0.00 | 0.69 ± 0.00 | 2.70 ± 0.00 |
| Wine | **0.05 ± 0.12** | 0.24 ± 0.01 | 0.21 ± 0.01 | 1.67 ± 0.14 |
| Yacht | **0.21 ± 0.10** | 0.39 ± 0.02 | 0.23 ± 0.05 | 1.29 ± 0.38 |

Table 3: Lower (↓) is better. Avg. NLL (± StdDev) of test examples on UCI datasets over 10 train-test splits.

## A.6 Density Estimation Visualization: Extended

In Figure 15, we present further results on density estimation with various decoder sampling techniques (top-$k$, top-$p$, low temperature) alongside MDN and Riemann baselines. We see that using vanilla temperature sampling for the decoder is optimal and unbiased for capturing the shapes of all problems.

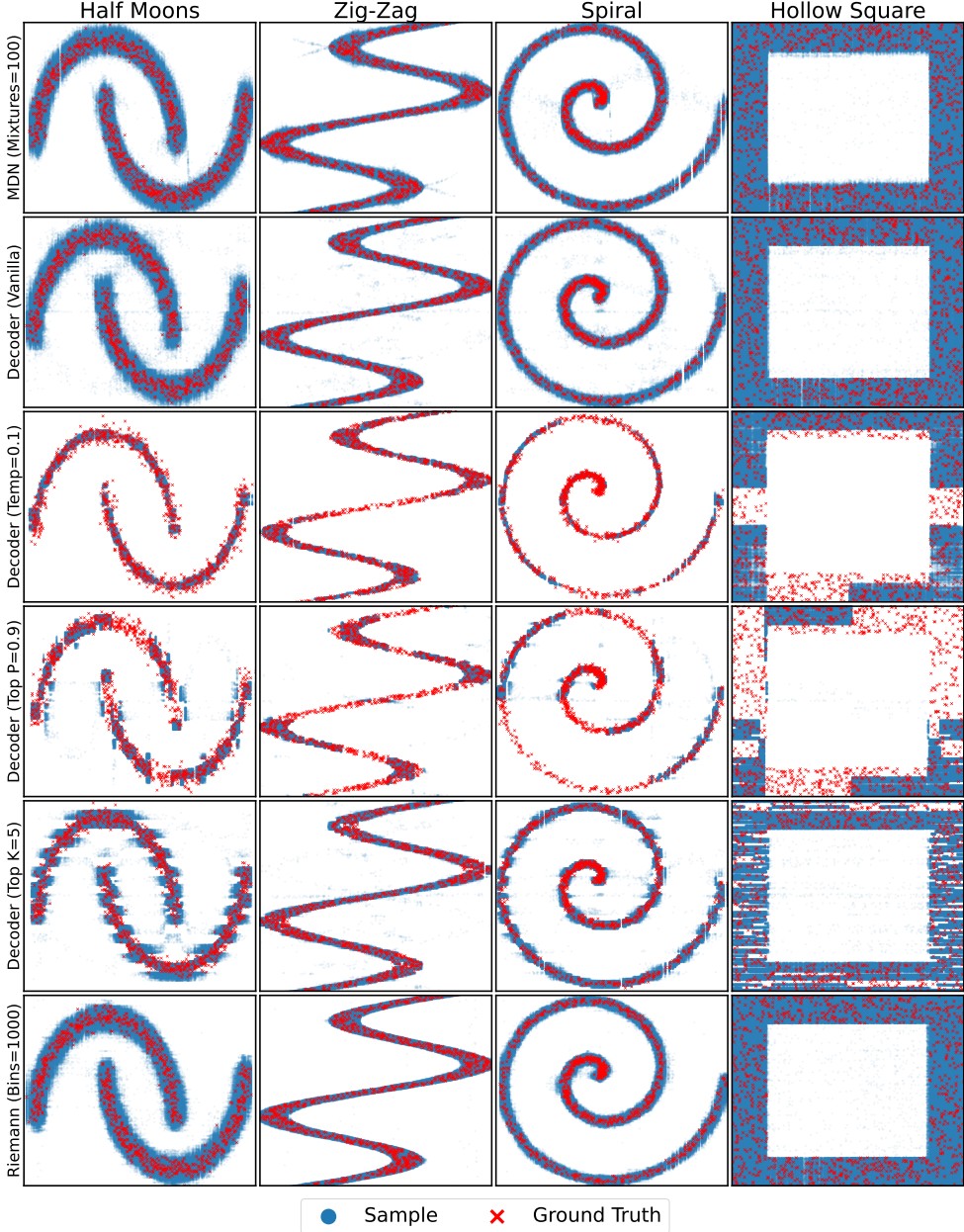

Figure 15: Visualizing density estimation of $p(y|x)$ on 1D problems. We used an unnormalized decoder with $(B = 10, E = 1, M = 5)$. Note that these results occur regardless of xy-scales, which are omitted for brevity.

## B  Extended Theory

*Proof of Theorem 1.* Firstly, we observe that

$$\underset{\theta}{\operatorname{argmin}} \frac{1}{N} \sum_{n=1}^{N} -\log p_{\theta}(\lambda_K(Y_n)) = \underset{\theta}{\operatorname{argmin}} H(\widetilde{f}_N^K, p_{\theta}),$$

where $\widetilde{f}_N^k$ is a discrete distribution that tracks the fraction of samples $\{Y_n\}_{n=1}^{N}$ that fall within each of the $2^k$ uniformly-spaced bins in $[0, 1]$. Formally,

$$\widetilde{f}_N^k((b_1, \ldots, b_k)) = \frac{1}{N} \sum_{n=1}^{N} \mathbf{1}(\lambda_k(Y_n) = (b_1, \ldots, b_k)).$$

Conditioned on the samples $\{Y_n\}_{n=1}^{N}$, $\widetilde{f}_N^k$ is a distribution on $k$-bit strings, and so by the $K$-bit universality assumption, $p_{\theta*}^K = \widetilde{f}_N^K$. It follows that $p_{\theta*}^k = \widetilde{f}_N^k \ \forall k \leq K$ since if two discrete distributions are equal so are any of their marginals. Then $f_N^{k*}(x) \equiv 2^k p_{\theta*}^k(\lambda_k(x)) = 2^k \widetilde{f}_N^k(\lambda_k(x))$ lines up exactly as a $2^k$-bin histogram estimator for $f$, for all $k \leq K$.

Now, we can treat the problem as one of histogram estimation. Let's consider a fixed $k$. We first observe that the risk can be written as the sum of a squared bias term and a variance one. Specifically,

$$R(f, f_N^{k*}) = \int_0^1 \operatorname{Bias}(y)^2 dy + \int_0^1 \operatorname{Variance}(y) dy,$$

where $\operatorname{Bias}(y) = \mathbb{E}[f_N^{k*}(y)] - f(y)$ and $\operatorname{Variance}(y) = \mathbb{V}(f_N^{k*}(y))$ is the bias and variance of $f_N^{k*}(y)$ at fixed $y$ respectively.

Now, label bins $\{B_j\}_{j=0}^{2^k-1}$, where $B_j = [j\varepsilon, (j+1)\varepsilon)$ and $\varepsilon = 2^{-k}$ is the bin width. Let $p_j = \int_{B_j} f(z) dz$ be the true probability mass in bin $B_j$. With $N_j$ as the number of samples in $B_j$, the expected value of the estimator for $y \in B_j$ is $\mathbb{E}[f_N^{k*}(y)] = \mathbb{E}[N_j/(N\varepsilon)] = (Np_j)/(N\varepsilon) = p_j/\varepsilon$.

Assume the true density $f$ is twice continuously differentiable on $[0, 1]$ (i.e., $f \in C^2([0, 1])$). This implies $f$, $f'$, and $f''$ are bounded on $[0, 1]$. Let $M_1 = \sup_{y \in [0,1]} |f'(y)|$ and $M_2 = \sup_{y \in [0,1]} |f''(y)|$.

**Bias Analysis:** Let $y_j = (j + 1/2)\varepsilon$ be the midpoint of bin $B_j$. For $z \in B_j$, by Taylor's Theorem around $y_j$: $f(z) = f(y_j) + (z - y_j)f'(y_j) + \frac{(z-y_j)^2}{2}f''(\xi_z)$ for some $\xi_z$ between $z$ and $y_j$. Integrating over $B_j$:

$$\begin{aligned}
p_j = \int_{B_j} f(z) dz &= \int_{B_j} \left[ f(y_j) + (z - y_j)f'(y_j) + \frac{(z-y_j)^2}{2}f''(\xi_z) \right] dz \\
&= f(y_j) \int_{B_j} dz + f'(y_j) \int_{B_j} (z - y_j) dz + \int_{B_j} \frac{(z-y_j)^2}{2}f''(\xi_z) dz \\
&= \varepsilon f(y_j) + 0 + R_j,
\end{aligned}$$

where the remainder term $R_j = \int_{B_j} \frac{(z-y_j)^2}{2}f''(\xi_z) dz$. Since $|z - y_j| \leq \varepsilon/2$ and $|f''(\xi_z)| \leq M_2$, we have $|R_j| \leq \int_{B_j} \frac{(\varepsilon/2)^2}{2} M_2 dz = \frac{M_2 \varepsilon^2}{8} \int_{B_j} dz = \frac{M_2 \varepsilon^3}{8}$. Thus, $R_j = \mathcal{O}(\varepsilon^3)$.

The bias for $y \in B_j$ is $\operatorname{Bias}(y) = \mathbb{E}[f_N^{k*}(y)] - f(y) = \frac{p_j}{\varepsilon} - f(y) = \frac{\varepsilon f(y_j) + R_j}{\varepsilon} - f(y) = f(y_j) + \frac{R_j}{\varepsilon} - f(y)$. Expanding $f(y)$ around $y_j$: $f(y) = f(y_j) + (y - y_j)f'(y_j) + \frac{(y-y_j)^2}{2}f''(\eta_y)$ for $\eta_y$ between $y$ and $y_j$.

$$\begin{aligned}
\operatorname{Bias}(y) &= f(y_j) + \mathcal{O}(\varepsilon^2) - \left[ f(y_j) + (y - y_j)f'(y_j) + \mathcal{O}(\varepsilon^2) \right] \\
&= -(y - y_j)f'(y_j) + \mathcal{O}(\varepsilon^2).
\end{aligned}$$

Now, integrate the squared bias over bin $B_j$:

$$
\begin{aligned}
\int_{B_j} \text{Bias}(y)^2 dy &= \int_{B_j} \left[ -(y - y_j)f'(y_j) + \mathcal{O}(\varepsilon^2) \right]^2 dy \\
&= \int_{B_j} \left[ (y - y_j)^2 (f'(y_j))^2 - 2(y - y_j)f'(y_j)\mathcal{O}(\varepsilon^2) + \mathcal{O}(\varepsilon^4) \right] dy \\
&= (f'(y_j))^2 \int_{B_j} (y - y_j)^2 dy - \mathcal{O}(\varepsilon^2)f'(y_j) \int_{B_j} (y - y_j) dy + \int_{B_j} \mathcal{O}(\varepsilon^4) dy \\
&= (f'(y_j))^2 \int_{-\varepsilon/2}^{\varepsilon/2} u^2 du - \mathcal{O}(\varepsilon^2)f'(y_j) \cdot 0 + \mathcal{O}(\varepsilon^4) \cdot \varepsilon \quad (\text{let } u = y - y_j) \\
&= (f'(y_j))^2 \left[ \frac{u^3}{3} \right]_{-\varepsilon/2}^{\varepsilon/2} + \mathcal{O}(\varepsilon^5) \\
&= (f'(y_j))^2 \frac{\varepsilon^3}{12} + \mathcal{O}(\varepsilon^5).
\end{aligned}
$$

Summing over all bins:

$$
\begin{aligned}
\int_0^1 \text{Bias}(y)^2 dy &= \sum_{j=0}^{2^k-1} \int_{B_j} \text{Bias}(y)^2 dy = \sum_{j=0}^{2^k-1} \left[ (f'(y_j))^2 \frac{\varepsilon^3}{12} + \mathcal{O}(\varepsilon^5) \right] \\
&= \frac{\varepsilon^2}{12} \sum_{j=0}^{2^k-1} (f'(y_j))^2 \varepsilon + \sum_{j=0}^{2^k-1} \mathcal{O}(\varepsilon^5) \\
&= \frac{\varepsilon^2}{12} \left( \int_0^1 (f'(y))^2 dy + \mathcal{O}(\varepsilon^2) \right) + 2^k \mathcal{O}(\varepsilon^5) \\
&= \frac{\varepsilon^2}{12} \int_0^1 (f'(y))^2 dy + \mathcal{O}(\varepsilon^4),
\end{aligned}
$$

where the third line uses a known approximation error for the Riemann sum with midpoint rule applied to $(f')^2 \in C^1$.

**Variance Analysis:** The variance for $y \in B_j$ is $\text{Variance}(y) = \mathbb{V}(f_N^{k*}(y)) = \mathbb{V}(N_j/(N\varepsilon)) = \frac{1}{(N\varepsilon)^2} \mathbb{V}(N_j)$. Since $N_j \sim \text{Binomial}(N, p_j)$, $\mathbb{V}(N_j) = Np_j(1 - p_j)$.

$$
\begin{aligned}
\text{Variance}(y) &= \frac{Np_j(1 - p_j)}{N^2 \varepsilon^2} = \frac{p_j(1 - p_j)}{N\varepsilon^2} \\
&= \frac{(\varepsilon f(y_j) + \mathcal{O}(\varepsilon^3))(1 - \varepsilon f(y_j) - \mathcal{O}(\varepsilon^3))}{N\varepsilon^2} \\
&= \frac{\varepsilon f(y_j) - \varepsilon^2 f(y_j)^2 + \mathcal{O}(\varepsilon^3)}{N\varepsilon^2} \\
&= \frac{f(y_j)}{N\varepsilon} - \frac{f(y_j)^2}{N} + \mathcal{O}(\varepsilon/N).
\end{aligned}
$$

Integrating the variance:

$$
\begin{aligned}
\int_0^1 \text{Variance}(y)dy &= \sum_{j=0}^{2^k-1} \int_{B_j} \left( \frac{f(y_j)}{N\varepsilon} + \mathcal{O}(1/N) \right) dy \\
&= \sum_{j=0}^{2^k-1} \left( \frac{f(y_j)\varepsilon}{N\varepsilon} + \mathcal{O}(\varepsilon/N) \right) \\
&= \frac{1}{N\varepsilon} \sum_{j=0}^{2^k-1} f(y_j)\varepsilon + 2^k \mathcal{O}(\varepsilon/N) \\
&= \frac{1}{N\varepsilon} \left( \int_0^1 f(y)dy + \mathcal{O}(\varepsilon^2) \right) + \mathcal{O}(1/N) \quad \text{(Riemann sum error for } f \in C^2) \\
&= \frac{1}{N\varepsilon} (1 + \mathcal{O}(\varepsilon^2)) + \mathcal{O}(1/N) \\
&= \frac{1}{N\varepsilon} + \mathcal{O}(\varepsilon/N) + \mathcal{O}(1/N).
\end{aligned}
$$

Since we typically consider asymptotics where $N \to \infty$ and $\varepsilon \to 0$ such that $N\varepsilon \to \infty$, the dominant variance term is $1/(N\varepsilon)$.

**Total Risk:** Combining the integrated squared bias and integrated variance:

$$
\begin{aligned}
R(f, f_N^{k*}) &= \int_0^1 \text{Bias}(y)^2 dy + \int_0^1 \text{Variance}(y)dy \\
&= \left( \frac{\varepsilon^2}{12} \int_0^1 (f'(y))^2 dy + \mathcal{O}(\varepsilon^4) \right) + \left( \frac{1}{N\varepsilon} + \mathcal{O}(\varepsilon/N) + \mathcal{O}(1/N) \right) \\
&= \frac{\varepsilon^2}{12} \int_0^1 (f'(y))^2 dy + \frac{1}{N\varepsilon} + \mathcal{O}(\varepsilon^4) + \mathcal{O}(1/N). \quad \text{(assuming } \varepsilon/N \text{ is smaller than } 1/N)
\end{aligned}
$$

Substituting $\varepsilon = 2^{-k}$:

$$
R(f, f_N^{k*}) = \frac{2^{-2k}}{12} \int_0^1 (f'(y))^2 dy + \frac{2^k}{N} + \mathcal{O}(2^{-4k} + 1/N).
$$

This gives the asymptotic risk. The $\mathcal{O}(2^{-4k} + 1/N)$ term is negligible, and can be disregarded. $\qquad \square$

## C    Exact Experimental Details

For all models, we swept the encoder (basic MLP with ReLU activation) by varying the number of layers within [2,3,4,5] and hidden units within [256, 512, 2048].

For $x$-normalization, we apply a mean and std scaling, i.e. $x \leftarrow (x - x_{mean})/x_{std}$ where $x_{mean}, x_{std}$ are coordinate-wise mean and standard deviations over all $x$'s in the training set. The preprocessed tensor is then fed directly into the encoder.

For $y$-normalization, we apply min/max linear scaling, i.e. $y \leftarrow (y - y_{min})/(y_{max} - y_{min})$ where $y_{min}, y_{max}$ are computed from the training set. This is applicable to models representing $[0, 1]$ output range (i.e. Riemann and Normalized Decoder). For Pointwise and Mixture Density heads, we further apply a shift $y \leftarrow y - 0.5$ to center the values within $[-0.5, 0.5]$.

All models were trained with a maximum of 300 epochs. To prevent overfitting, we apply early stopping (patience=5) where the validation split is 0.1 on the training set. Adam learning rates were swept over [1e-4, 5e-4].

We further describe hyperparameters and sweeps for individual heads below:

**Pointwise:** Uses ReLU activations on every hidden layer.

- Weight decay: [0.0, 0.1, 1.0]

**Unnormalized Decoder:** Uses vanilla temperature sampling.

- Base $B$: [4, 8, 10]
- Exponent Digit Count $E$: [1, 2, 4]
- Mantissa Digit Count $M$: [2, 4, 8]
- Transformer size: (3 layers, 128 units, 4 heads) or (1 layer, 32 units, 1 head).

**Normalized Decoder:** Sampling same as unnormalized decoder.

- Base $B$: [2, 4, 8]
- Length $K$: [4, 8, 6]
- Transformer size: Same as unnormalized decoder.

**Riemann/Histogram Distribution:** We specify a bin count, which uniformly partitions the range $[0, 1]$ into equally spaced bins. Output is parameterized using softmax.

- Bin Count: [16, 64, 256, 1024, 4096, 16384]

**Mixture Density Network:** Given a mixture count $M$, the distribution head consists of mixture $\pi_M \in \triangle^M$, mean $\mu_M \in \mathbb{R}^M$, and standard deviation $\sigma_M \in \mathbb{R}^M$. Mixtures were parameterized using softmax, while standard deviations were via $\text{ELU}(x) + 1$ activation to enforce positivity.

- Mixtures $M$: [1, 2, 5, 10, 20, 50, 1000]

