# OpenReview forum: "Decoding-based Regression"
_TMLR — Accepted by TMLR_

### Review · Reviewer_zAuL · 2025-06-10

**Summary Of Contributions:**

This work investigates the capabilities of autoregressive models to perform regression tasks based on tokenized target variables. Specifically, this work provides an approximation bound between  distributions with twice continuously differentiable density functionsthe first K bits and their maximum likelihood K-bit approximation.
Moreover, such models are empirically evaluated on regression and density estimation tasks.

**Audience:**

Yes

**Broader Impact Concerns:**

I don't have concerns on the ethical implications of this work.

**Claims And Evidence:**

No

**Requested Changes:**

## Critical
**Clarity.** The formal framework and the experimental setting needs to be described clearly.

**Soundness.** The empirical comparison needs to be extended such that it sufficiently supports the claim *"decoding-based regression is as performant as traditional approaches when benchmarked over standard regression tasks."* Statements regarding fitting arbitrary distributions need to be made more precise.

## Minor
The readability of Theorem 1 would improve if definitions are moved outside.

A figure that illustrates the architecture and the relation between individual components (encoder, decoder, auto-regressive model / transformer, heads) would be appreciated.

Typos:
"**We** call **the** parametric model...
"Define **the** risk R as the mean integrated squared error between **the** true ..."
"One can also improve regression behavior using techniques purely by modifying sequence representations."

**Strengths And Weaknesses:**

# Strengths
- This work provides both, theoretical and empirical, support for its claims.
- The experiments use a wide range of datasets, which suggests that their findings are universal and not constrained to the considered data.
- The proof is a bit hard to read, but seems to be correct. However, I have not checked it thoroughly.

# Weaknesses
The most severe weakness is clarity. I had trouble understanding the theoretical framework, the considered neural network architecture and the setup of the experiments. In consequence, I am unable to verify the soundness of the current version of this submission.
Specifically, the following points need more explanation.

  - Key concepts of the considered architecture remain unclear. Presumably, the model architecture consists of three parts, an autoregressive model operating on the inputs, an encoder that tokenized the response variable and the head which produces the prediction of the response variable. This is not clearly specified though, so I am unsure if my description is accurate and precise enough.
  Specifically, the key component of the network that is studied in this work, i.e., the head is not clearly specified, and its unclear to me, what exactly is meant by a regression head, a decoder head, a pointwise head and a Riemann head.
  - Related to the previous point is that since I do not know what the different heads are, I also do not know what exactly is evaluated in the experiments. For example, I wonder if the tokenization of the response variable is necessary to achieve a good regression or if keeping is as a float leads to similar results. Maybe this is already studied in the experiments, but I cannot tell.
  - The theorem statement is difficult to decipher.
    - From my understanding, it seems, that the theorem is actually a result about the mse of the maximum likelihood estimation of the first K bits of some random variable, and that everything model specific can be neglected due to the K-bit universality assumption. Is this correct?
    - What is the relation between $\hat \theta$ and $\hat f_N$. In which part is it relevant, that $\hat \theta$ is a maximum likelihood estimator?
    - Later on, it is stated that *"We can decompose this [K-bit universality] assumption further into two pieces: 1) that there exists θ∗ in our model class that achieves the minimum cross-entropy (i.e. pθ∗ = p in Definition 1), and 2) that our SGD-based training procedure is able to find it."* However, it seems that K-bit universality only concerns part 1).
    - More generally speaking, what is the interpretation of the Theorem and why is it relevant to decoding-based regression?

  - In the preliminaries, the tokenization is described for arbitrary base B, but the theory is limited to base 2 (bits). I therefore wonder whether the choice of base 2 is for simplicity only, or if it is necessary.
  - At the start of section 4 (Experiments) it is written that *"This minimizes the decoding head’s contribution to representation learning (as opposed to numeric modeling), since it will be relatively negligible in size to ϕ(x), i.e. less than 10% of the total network parameter count."*
  What do you mean with numeric modeling here and why do we want to minimize the decoders contribution?

Regarding the experiments.
- Comparisons with baselines are missing. This includes a comparison to standard baselines such as tree based and kernel methods, as well as neural networks other than transformers.
- In figure 2, what is meant with 'theoretical', 'decoder', 'Riemann'. How are the risk values obtained?
- Figure 6 is missing labels.
- What is meant with 'vanilla temperature sampling'?

In the abstract, the following claim is made *"We find that, despite being trained in the usual way - for next-token prediction via cross-entropy loss - decoding-based regression is as performant as traditional approaches when benchmarked over standard regression tasks, while being flexible enough to capture arbitrary distributions, such as in the task of density estimation."*
This statement lacks sufficient support, as (i) there is no empirical comparison to *"traditional approaches"* and the theory only concerns specific distributions, i.e., targets are scalar and have a C2 density.

---

> ### Author Response · Authors · 2025-06-10
> **Response to Reviewer zAuL**
>
> Hi Reviewer zAuL, thank you for the comprehensive reading. We've made edits to the updated draft in **blue**.
>
> Direct clarifications and replies below:
>
> ## Presentation and Theory
> * **Considered architecture:** We assume the encoding $\phi(x)$ is already defined, specific to the regression task. E.g. for tabular regression, $x$ is first feature-engineered and $\phi$ is an MLP. For image regression, $x$ is an image and $\phi$ is a CNN. For LLM reward modeling, $x$ is a string and $\phi$ is a transformer. Our paper is investigating the output head applied on top of $\phi(x)$, so we aren't constrained to any particular $\phi(x)$ - we just assume it's a vector representation.
> * **Different Heads:** Updated draft to be more explicit about defining different regression heads. For your convenience, here are the heads:
>     * General Regression Head: Any post-processing applied to return numeric predictions.
>     * Pointwise: Deterministic projection of $\phi(x)$ to a scalar output.
>     * Riemann: Represents a histogram-like distribution parameterized by softmax over categories representing number bins. E.g. a softmax over bins <0>, <1>, ..., <999> to represent the [0, 1] interval using 1000 bins.
>     * Decoder: Uses autoregression to decode tokens representing a number. e.g. decoding three tokens <1><5><2> to represent the number 152.
> * **Theory:**
>     * Yes, the K-bit assumption just means that the head can perfectly fit any distribution on the K-bit support. This is a standard assumption (akin to universal neural network approximation) in order to say anything theoretically meaningful.
>     * Sorry for the confusion. We realized $\widehat{f}$ is overloaded notation for both MLE and estimator. We use $\theta^\*$ and $ f^\*$ to denote MLE now.
>     * Correct, K-bit universality only means there exists a perfectly-fitting argmax in the loss landscape, but doesn't say anything about how to get there. To actually get there by training, we'd also have to assume SGD is able to do so, which might be not true empirically. The empirical findings show implicit regularization is occurring on the decoder, maybe e.g. due to SGD-transformer interactions biasing toward smoother optima. Clarified in draft.
>    * The theorem is showing that _if a model with too high of a precision fits the training data perfectly, it can lead to overfitting_, and this leads to a predictable risk, shown precisely by the Riemann case. The decoder doesn't follow this overfitting trend, which allows it to remain highly numerically precise even in low data regimes. This suggests the decoder has nice implicit regularization properties which make it much more efficient with training data.
> * **Base B Necessary:** Yes, the proof for base 2 is for simplicity only. We can easily change the base, and the proof would still hold, after changing the constants in the final risk calculation (e.g. $2^k$ changes to $B^k$, $12$ changes to another constant, etc.)
> * **Negligible Decoder:** A regressor's performance is determined by two factors, its feature encoding and its output head. For apples-to-apples comparisons on regression heads, we ideally need to keep the feature encoder (i.e. the MLP) the same throughout all experiments. We can approximately achieve this effect by making the MLP so big, that the regression head's parameters' contribution to feature encoding is negligible.
>
> ## Experiments:
> * **Comparisons with tree/kernel baselines:** These are not deep-learning based and would be red herrings since they don't affect the conclusions (comparison between different deep-learning regressor heads, and the justification of decoder heads). Note that we're only using tabular benchmarks just for convenience -- our focus isn't really about tabular tasks at all either, and we don't claim anything specific about SOTA on tabular regression. We've updated the abstract (decoder-based heads vs standard pointwise heads) to be more specific on our contribution.
> * **Figure 2 labels:** The theoretical follows directly from Theorem 1. Decoder and Riemann values are obtained experimentally,by trying to fit an example Gaussian distribution.
> * **Figure 6 labels:** Added task indices.
> * **Vanilla Temperature Sampling:** In the decoder head, vanilla temperature sampling is a common phrase, just meaning that each token is sampled with basic softmax (no special methods like TopP, TopK, Nucleus, etc. used for LLMs).
>
> ## Abstract
> We mean "traditional approaches" simply as traditional regression heads (e.g. a MLP with a pointwise scalar output is a standard baseline in regression). By "arbitrary distributions", we meant arbitrary floating point distributions (in the context of regression). We modified the abstract to be more precise on our claims, i.e. "decoder-based heads are as performant as standard pointwise heads when benchmarked over standard regression tasks, while remaining flexible enough to capture smooth numeric distributions".

---

> > ### Comment · Reviewer_zAuL · 2025-06-23
> > **Follow up questions**
> >
> > Thank you for your response. I looked at the revision, and while it is certainly an improvement, my concerns have not been alleviated so far.
> >
> > Being more explicit about the definition of the heads certainly helps. However, the issue still remains, as the terms regression head, Gaussian distribution head, decoding based regression head are still used before they are properly defined (if they are). In fact, there is even more terminology now (feed-forward regression head, parametric distribution head, sigmoid head, decoding head vs decoding-based regression head, trainable head)
> >
> > To be clear, if you say head, do you mean the last layer (plus potentially a subsequent activation / normalization function) of a neural network? If so, do you require this layer to be linear?
> > If not, what do you mean instead? And how do you determine which layers are part of the head and which layers are part of the encoder, i.e., which part of the network is responsible for obtaining good representations and which part is responsible for the regression or density estimation?
> >
> > In the experiments, do you always use the same encoder $\phi$ for the different heads, i.e., $\phi$ is pretrained and then frozen during the training of the heads. Again, if not, how do you determine which part of the network is responsible for obtaining good representations and which part is responsible for the regression?
> >
> > Where does the decoder fit into this description?  (Please stress in the paper, that despite common terminology in the context of variational autoencoders, you do not consider distribution heads to be decoders)
> > Where does the transformer model fit into this description? Is it the decoder? If so, what is a transformer with "only 1 layer and 32 units"?
> >
> > I also wonder, why only the pointwise heads are considered tensor based. Their output is a scalar, and the input is the same as the input to the other heads (since you seem to always use the same encoder architecture, a large MLP), i.e., a vector or tensor.

---

> > > ### Author Response · Authors · 2025-06-23
> > > **Update**
> > >
> > > Hi Reviewer zAuL, we've made additional changes based on your follow-up comments:
> > > * Section 2, we've re-organized the heads. Now "Pointwise Heads" only refer to "deterministic learnable functions (with weights $\theta$), typically a simple feed-forward network (often a single layer) mapping $phi(x)$ to a single-point scalar."
> > > * There is now a separate paragraph on "Parametric Distribution Heads" (e.g. single Gaussian or Gaussian mixtures).
> > > * Section 3, we give a brief summary paragraph on the decoder head, "autoregressive sequence model, such as a compact Transformer decoder...".
> > > * Section 3, we also give a note that our use of the term "decoder" in the context of language modeling is not to be confused with VAEs.
> > >
> > > To directly answer your questions:
> > > * Definition of a regression head: Notationally, we can define a regression head, denoted as $p_{\theta}(y | \phi(x))$, is a subsequent module that takes the feature representation $\phi(x)$ as input and produces the final numeric prediction.
> > >     * It is true that for a neural network, the split between the head and the features can be defined arbitrarily anywhere, and we acknowledge this imprecision. Yes, the _canonical_ way is to define the head as the very last layer (plus activation/normalization), but it's possible that e.g. in reinforcement learning (RL), people define "policy head" and "value head" as having a few more additional MLP layers.
> > > * In our experiments, we acknowledged and dealt with this imprecision issue by making the feature encoder $\phi(x)$ _much bigger_ (i.e. tuning up to 5 ReLU layers, each with up to 2048 units), while ensuring the "head" is much smaller (less than 10% of total parameter count). This strongly minimizes the effect of the "head" for any feature processing. Specifically for different heads:
> > >   * Pointwise and Gaussian heads used linear projections right after the MLP (so the linear weight shapes are at most 2048 x 1)
> > >   * Decoder Transformer head used 1 layer and 32 units.
> > > * Frozen vs. trainable $\phi$: For our experiments, we made $\phi$ trainable for the sake of simplicity, but in other applications (e.g. $\phi$ is a LLM embedding), it can be frozen (but differentiable through) or over-the-wire (e.g. pinging ChatGPT service)
> > > * Decoder terminology: While we've added the VAE note in Section 3, overall in the current LLM era, we believe "decoder" now commonly refers to language modeling and Transformers. In addition, our abstract's first phrase is "language model", which should set the context.
> > > * Pointwise heads considered tensor-based: Changed; we no longer use the term "tensor-based" for clarity.

---

> > > > ### Comment · Reviewer_zAuL · 2025-06-24
> > > >
> > > > The consistent use of decoder **head** and the updates to section 2 in the current revision make things much clearer now.
> > > > But please stress in the paper, that despite common terminology in the context of variational autoencoders, you do not consider distribution heads to be decoders (the added note misses the point of why there even might be confusion).

---

> > > > > ### Author Response · Authors · 2025-06-24
> > > > > **Updated**
> > > > >
> > > > > Understood - at the end of Section 2, we now make the distinction in blue, based on output spaces:
> > > > > * To prevent confusion with the term 'decoder' which is also a central component of generative models like Variational Autoencoders (VAEs) (Kingma & Welling, 2014), we stress a key distinction. While both VAE decoders and distributional regression heads map a feature vector to a probability distribution, their objectives differ: a VAE decoder models $p(x|\phi(x))$ to reconstruct the input $x$ from a latent $\phi(x)$, whereas a regression head models $p(y|\phi(x))$ to predict a separate target y. Due to this difference in output space, we do not refer to standard distributional heads (e.g., Gaussian) as 'decoders'."
> > > > >
> > > > > Please let us know if there are more concerns.

---

> > > > > > ### Author Response · Authors · 2025-06-30
> > > > > > **Rebuttal Period Ending**
> > > > > >
> > > > > > Hi Reviewer zAuL, it seems that the rebuttal period is close to ending (July 2).
> > > > > >
> > > > > > Please let us know if there are any additional concerns or issues we can address, or if our updates have changed your mind on and claims and evidence of the paper.
> > > > > >
> > > > > > Thanks!

---

### Review · Reviewer_hcDX · 2025-06-12

**Summary Of Contributions:**

This paper examines the use of token-based autoregressive decoders for regression or density estimation tasks. The authors propose two tokenization schemes, one for normalized data and one for unnormalized values. In the first case, the model predicts a length-K sequence of base-B tokens $t_{1..k}$, approximately corresponding to a value of $\sum_{i=K}B^{-i}t_i$. For the second case, the tokenization scheme generalizes floating point approximations, where the model predicts the sign, mantissa and exponent. Experimental results show that such token-based auto-regressive decoder heads perform similarly or better than pointwise scalar regression heads.

**Audience:**

Yes

**Broader Impact Concerns:**

None.

**Claims And Evidence:**

Yes

**Requested Changes:**

**Would strengthen the work**

- Page 3, section 3.2. You mention that "Since the token space is finite while $\mathbb{R}$ is uncountable, this mapping is lossy". The mapping would also be lossy for countably infinite sets.
- Page 3, section 3.2. You mention "Any autoregressive model can be used, so long as it supports constrained token decoding to
enforce proper sequences which represent a valid number.". If you disable constrained decoding, how often does the model generate a non-valid output?
- Page 11, section 4.5. How does predicting the output multiple times sequentially compare to predicting it in parallel? The latter wouldn't require extending the training data.

**Strengths And Weaknesses:**

**Strengths**

- The paper proposes simple and well-principled approaches, which are supported by some theoretical foundation, to use token-based autoregressive decoders for regression tasks.
- Results show that such approaches are competitive (and sometimes superior) to traditional baselines. The figures also convincingly show the effectiveness of the proposed solutions.

**Weaknesses**

- While the paper is motivated via language modeling capabilities, the decoder heads do not really model language anymore due to the choice of tokenization schema. I would be curious to know if the approach could be extended to support regression and language modeling jointly, either by reserving a set of tokens for regression, or re-using some tokens for both tasks.

---

> ### Author Response · Authors · 2025-06-13
> **Response to Reviewer hcDX**
>
> Hi Reviewer hcDX, thank you for your kind words and positive review, and we're especially glad that you understood our work and purpose very well. Responses below:
>
> * **Decoder head vs language models:** We believe our decoder head is a simplified but instructive demonstration of what happens with LLMs, across all aspects (tokenization, training, inference):
>    * For tokenization, modern LLMs now tokenize numbers digit-by-digit, so if Base=10, the normalized scheme is analogous to representing a number as e.g. "0.1234", while the unnormalized scheme is equivalent to scientific notation, e.g. $10^{-4} \times 1234$.
>   * Our training procedure is also the same as regular LLMs (i.e. cross-entropy loss on tokens).
>   * For inference, as we also answered in another question, our constrained decoding is not absolutely necessary - TopK or TopP sampling (standard for LLMs) can also be used instead, to zero-out wrong token probabilities. We ablated these sampling methods in Appendix A5.
> * **Uncountable:** Apologies - did you mean $\mathbb{R}$ (used in our paper) instead of $\mathbb{Q}$ (typo)? In any case, we agree there would also be a lossy mapping if our output space was only $\mathbb{Q}$, and it is exciting that decoding-based regression makes such exotic regression tasks much easier.
> * **Disabling constrained decoding:** Great question. The issue stems from numerics - i.e. softmax (with temperature=1) simply doesn't lower enough the probability of choosing a "wrong" token during decoding, and as long as this probability is non-zero, a failure could still happen. Constrained decoding isn't the only solution - standard sampling strategies such as TopK or TopP can also be used to zero-out wrong token probabilities.
>   * For example, we trained our decoder on 20K examples from a synthetic regression task, and still found wrong tokens during sampling, since the probability of choosing e.g. a sign token for the mantissa still remained ~ 10^{-4} to 10^{-5} probability.
> * **Predicting in parallel vs sequentially:** We believe these are orthogonal methods (sequential vs parallel) - for the majority of the paper we already performed parallel sampling to better estimate the underlying distribution $p_{\theta}(y|x)$, and saw that pointwise errors reduced up until 256 samples, from which improvements were very marginal. Sequential sampling offers additional error reduction benefits and possesses a different performance scaling (e.g. up to 5 sequential repeats).
>
> Thanks in advance, and let us know if more questions arise! -- Authors

---

### Review · Reviewer_tEwp · 2025-06-18

**Summary Of Contributions:**

This paper investigates the performance of autoregressive models on regression tasks, with a particular focus on different choices for the final layer ("head") of the network. It provides theoretical justification for using decoder-based heads in this setting, including a result that bounds the error of decoder-based regression. Finally, experiments demonstrate that standard ("pointwise") heads achieve performance comparable to that of decoder-based heads.

**Audience:**

Yes

**Claims And Evidence:**

Yes

**Requested Changes:**

In my view, the paper requires moderate to substantial rewriting before it can be published. In particular, I identified the following areas where the presentation can be improved:

- Abstract: The terms "prediction heads" and "decoder-based heads" are not standard. While they are defined later on page 2, their use in the abstract may confuse readers. Moreover, the word "head" is commonly used to refer to attention heads, so it would be helpful to clearly distinguish the two early in the text.
- The Introduction should more clearly state the contributions of the paper. As it stands, it is unclear whether the paper is primarily theoretical or empirical, and what its main strengths are.
- pg. 2: The sentence "However, we argue that this approach ... works." is unclear.
- Section 3.2: The statement "If y is restricted to [0, 1] (via scale normalization for example), then we show we can..." should specify where this result is shown.
- Figure 5, caption: Phrases like "Fit to ground truth is better" and "practically unlimited $(x, y)$ points" should be replaced with precise, formal statements.

**Strengths And Weaknesses:**

**Strengths**

The paper studies an interesting problem: the application of autoregressive models to regression tasks (predicting a single real value). It provides valuable insights into a particular element of the regression pipeline: the final layer of the architecture and the tokenization scheme. Several interesting experiments on both real and synthetic data illustrate the methods considered. The theoretical contribution appears to be correct.

**Weaknesses**

Despite the nice contributions of the paper, its presentation is flawed. Several parts of the paper use imprecise language, making it difficult to evaluate the correctness of some claims. See examples in the Requested Changes section.

---

> ### Author Response · Authors · 2025-06-18
> **Response to Reviewer tEwp**
>
> Hi Reviewer tEwp - thanks for the kind comments on our paper's valuable insights and interesting results. We've modified the wording (highlighted in blue) in the updated draft to reflect your requested changes.
>
> Specifically:
> * "prediction head" - To the best of our ability, we made wording changes so that "head" is more obvious. The abstract now begins with "numeric regression heads", and the intro also mentions it is the modeling of $p(y|x)$. We believe that the term "regression head" is common, as a [Google Search](https://www.google.com/search?q=regression+head) leads to numerous results.
> * We've reworded the introduction. Now, it states:
>     * "We formalize decoding-based regression, i.e. explicitly define tokenization schemes for numbers, establish training and inference procedures, discuss methods for pointwise estimation, and theoretically provide risk guarantees for density estimation under common assumptions."
>     * "In experimental benchmarks, properly tuned decoding-based regression heads are data-efficient and competitive with regular pointwise heads on tabular regression, yet are also expressive enough to perform against Gaussian mixture heads for density estimation."
> * "However we argue that this approach" -> "we argue that token-based numeric modeling"
> * Section 3.2: We've referenced the theory in Section 3.4
> * Figure 5: Changed to "Visual fit to ground truth is better" and "practically unlimited" -> "100K"
>
> We made some further small adjustments to the wording throughout the paper. Overall, we've tried our best to edit any ambiguities, but we believe that these are not fatal to our fundamental claims and evidence, but rather small grammar or phrasing changes. If the Reviewer agrees that we now clearly back up our core claims with sufficient evidence, we kindly ask that the Reviewer change their score.
>
> Thanks in advance! Sincerely, Authors

---

> > ### Comment · Reviewer_tEwp · 2025-06-23
> >
> > Hi,
> >
> > I read the revised paper. I agree that the proposed changes improve the clarity and readability of the work (especially those in Section 2). I changed my score appropriately.

---

> > > ### Author Response · Authors · 2025-06-23
> > > **Thank you!**
> > >
> > > Thank you very much for your feedback and score change!

---

### Author Response · Authors · 2025-06-19
**General Comment to all Reviewers**

Hi Reviewers - we thank each and every one of you for your time and effort in reading our paper and providing careful feedback.

We've made very big edits to the overall readability of the paper (in **blue**), and would greatly appreciate it if you could skim the paper one last time during the rebuttal period.

We believe we've greatly improved the paper and handled all questions and concerns (either through paper editing or directly responding in OpenReview).

We hope that this changes some Reviewers' minds on our "Claims and Evidence" for the better, and if so, kindly and respectfully request if our score may be updated.

if you have any more questions - please let us know

Thanks for your time!

---

### Decision · Action_Editor_NhjM · 2025-07-26

**Recommendation:** Accept with minor revision

**Additional Comments:**

The initial reviews were mixed. While the reviewers acknowledged the appeal of token-based regression, the value of the theoretical risk analysis, and the promising empirical results, they also had significant reservations, in particular regarding the manuscript’s presentation and clarity. Major concerns included the paper’s imprecise language and vague/incorrect statements, as well as the implications of the theoretical contributions. During the discussion phase the authors revised the manuscript and — over multiple iterations — were able to address many of these concerns. Ultimately, all three reviewers recommended accepting the paper or are leaning this way.

While the paper weakly meets the general acceptance criteria (‘audience’, ‘claims and evidence’), there is still room for improvement and I urge the authors to continue their work on the paper’s presentation and clarity. Additionally, I want to highlight a few issues in the experiments that need to be addressed before the paper can be accepted:
- The experiments are based on very simple architectures, all of which are far away from state-of-the-art. It is therefore not clear if and to what extend the reported results would translate to modern LLMs. Please include a detailed discussion of this point, so that the reader has the necessary context.
- Please include two decimals in Table 1.
- For consistency, and using the same data:
	- please add “Normalized Decoder” and “Riemann” to all plots in Figure 4.
	- please add “Normalized Decoder” and “Riemann” to all plots in Figure 5.
- For completeness, please produce Figure 6 for the same AMLB tasks as shown in Figure 5 (potentially moving some of these results to the appendix) and explain why “Unnormalized Decoder” is not included in these experiments.

Once these changes have been made, the paper will be recommended for acceptance.

**Audience:**

Yes

**Audience Explanation:**

The paper explores the use of auto-regressive token prediction for real-valued regression tasks. To this end, the paper introduces two tokenization schemes — normalized and unnormalized — representing real numbers as a sequence of tokens. A theoretical analysis based on universality principles enables an exact description of the risk under this framework. Assuming an arbitrary feature representation, the experiments compare this token-based prediction to traditional point-, histogram-, and mixture-based approaches.

Due to the simplicity of the underlying architectures, some of the insights and conclusions reported in this paper are not directly transferable to state-of-the-art LLMs. Despite this, token-based regression is an appealing concept and at least some members of the community will thus be interested in the presented theoretical and empirical analyses.

**Claims And Evidence:**

Yes

**Claims Explanation:**

The proposed framework is supported by theoretical insights and empirical evidence. The experiments compare decoding-based regression to a number of more traditional output spaces, including point estimates, probability distributions, and histogram-based representations — both in synthetic and real-world scenarios. Ablation studies explore the size of the decoding head and the use of error correction.

---

> ### Author Response · Authors · 2025-07-31
> **Updated Camera Ready**
>
> Hi Action Editor, thanks so much for your comprehensive reading and follow-up editing suggestions. We have addressed all of them, specifically:
>
> * Additional Section 5 for "Discussion: Limitations and Extensions" around how the results may / may not translate to modern LLMs.
> * Two decimals in Table 1
> * Consistently for figures:
>     * All four regression heads are now displayed in Figure 4, along with edited analysis. Unnormalized decoder still performs the best, while the Riemann head struggles.
>     * Figure 5 points to full results in Appendix A3, Figure 13, containing all task performances over all four regression heads now. We've also added Figure 6 to specifically compare decoders vs. Riemann, and find decoders almost always beat the Riemann head.
> * Figure 7 (formerly referred as Figure 6) now points to Appendix A1, Figure 11 for full results. We omit unnormalized decoder results, as it aggregates samples differently from the other heads (which use mean aggregation) and would not be appropriate for a more rigorous comparison.
>
> Thanks so much!